# AAV-mediated editing of PMP22 rescues Charcot-Marie-Tooth disease type 1A features in patient-derived iPS Schwann cells

Yuki Yoshioka [1,7], Juliana Bosso Taniguchi [1,7], Hidenori Homma [1,7], Takuya Tamura [1], Kyota Fujita [1], Maiko Inotsume[1], Kazuhiko Tagawa[1], Kazuharu Misawa [2,3], Naomichi Matsumoto[2], Masanori Nakagawa[4], Haruhisa Inoue [5,6], Hikari Tanaka[1✉] & Hitoshi Okazawa [1✉]

## Abstract

**Background** Charcot-Marie-Tooth disease type 1A (CMT1A) is one of the most common hereditary peripheral neuropathies caused by duplication of 1.5 Mb genome region including *PMP22* gene. We aimed to correct the duplication in human CMT1A patient-derived iPS cells (CMT1A-iPSCs) by genome editing and intended to analyze the effect on Schwann cells differentiated from CMT1A-iPSCs.

**Methods** We designed multiple gRNAs targeting a unique sequence present at two sites that sandwich only a single copy of duplicated peripheral myelin protein 22 (*PMP22*) genes, and selected one of them (gRNA3) from screening their efficiencies by T7E1 mismatch detection assay. AAV2-hSaCas9-gRNAedit was generated by subcloning gRNA3 into pX601-AAV-CMV plasmid, and the genome editing AAV vector was infected to CMT1A-iPSCs or CMT1A-iPSC-derived Schwann cell precursors. The effect of the genome editing AAV vector on myelination was evaluated by co-immunostaining of myelin basic protein (MBP), a marker of mature myelin, and microtubule-associated protein 2(MAP2), a marker of neurites or by electron microscopy.

**Results** Here we show that infection of CMT1A-iPS cells (iPSCs) with AAV2-hSaCas9-gRNAedit expressing both hSaCas9 and gRNA targeting the tandem repeat sequence decreased *PMP22* gene duplication by 20–40%. Infection of CMT1A-iPSC-derived Schwann cell precursors with AAV2-hSaCas9-gRNAedit normalized *PMP22* mRNA and PMP22 protein expression levels, and also ameliorated increased apoptosis and impaired myelination in CMT1A-iPSC-derived Schwann cells.

**Conclusions** In vivo transfer of AAV2-hSaCas9-gRNAedit to peripheral nerves could be a potential therapeutic modality for CMT1A patient after careful examinations of toxicity including off-target mutations.

## Plain Language Summary

Charcot-Marie-Tooth disease type 1A (CMT1A) is a common heritable form of the condition that develops when nerves in the body's extremities, such as the hands, feet and arms, are damaged due to an extra copy of *PMP22* gene being incorrectly produced. Currently, no known therapies exist. Here, we developed a method to delete the additional copy of *PMP22* gene by 20–40% to prevent overproduction. Our results show that this method can reduce PMP22 protein production, leading to near normal production in patient's nerve cells. Further safety assessments should now be undertaken. If the treatment is safe for patients it could become a therapeutic option for CMT1A patients.

[1] Department of Neuropathology, Medical Research Institute, Tokyo Medical and Dental University, 1-5-45, Yushima, Bunkyo-ku, Tokyo 113-8510, Japan. [2] Department of Human Genetics, Yokohama City University Graduate School of Medicine, Yokohama, Kanagawa 236-0004, Japan. [3] RIKEN Center for Advanced Intelligence Project, 1-4-1 Nihonbashi, Chuo-ku, Tokyo 103-0027, Japan. [4] Department of Neurology, Kyoto Prefectural University of Medicine, Kyoto 606-8507, Japan. [5] Center for iPS Cell Research and Application (CiRA), Kyoto University, Kyoto 606-8507, Japan. [6] Drug-discovery cellular basis development team, RIKEN BioResource Center, Kyoto 606-8507, Japan. [7] These authors contributed equally: Yuki Yoshioka, Juliana Bosso Taniguchi, Hidenori Homma. ✉email: tanknpat@tmd.ac.jp; okazawa-tky@umin.ac.jp

Charcot-Marie-Tooth disease (CMT) is a group of familial neurodegenerative diseases that cause hereditary motor and sensory neuropathy (HMSN), with an incidence of 10–80 out of every 100,000 individuals[1–5]. More than 30 autosomal dominant genes are considered causative for CMT, and other causative genes are autosomal recessive or X-linked. The mutations cause axonal or demyelinating peripheral neuropathy[6–11]. Charcot-Marie-Tooth type 1 (CMT1) is the most common disease type, accounting for approximately 70% of CMT patients, and is caused by demyelinating peripheral neuropathy. Six subtypes of CMT1, CMT1A-F, are associated with different causative genes. CMT1A, which is caused by pathological duplication of the peripheral myelin protein 22 (PMP22) gene and subsequent accumulation of PMP22 protein, is the most common subtype, accounting for 70–80% of CMT1 patients[6–11]. Characteristic subjective symptoms and objective findings include distal muscle weakness, paresthesia, and mild impairment of peripheral nerve conduction velocities. The severity of symptoms is variable between cases[6–11].

Presently, no therapeutic interventions are available whose effects are clinically confirmed for CMT or CMT1A[6,9,10]. Primary management includes physiotherapy and occupational therapy and in some cases non-steroidal anti-inflammatory drugs (NSAIDs) or antidepressants to manage pain[6–10,12]. PMP22 down-regulation is one of the most promising therapeutic approaches for CMT1A. Small interfering RNA (siRNA)[13,14], AAV-mediated small hairpin inhibitory RNA (shRNA)[15] and AAV-mediated microRNA (miRNA)[16] have been developed at the research level to suppress abnormally increased PMP22 mRNA in CMT1A.

There are a number of mouse models of CMT1A[9,10,14,17–20] to examine the idea. However, they are transgenic mouse or rats overexpressing PMP22, and there is no knock-in animal model possessing PMP22 gene tandem duplication with sufficiently long flanking genome regions. A recent report[15] examined the effect of a shRNA to knock down PMP22 on a rat CMT1A model in which PMP22 mRNA expression is more than four folds[17]. Another recent work developed AAV-mediated expression of miRNA to down-regulated PMP22 with the heterozygous human PMP22 transgenic C61 mice in which PMP22 mRNA expression is increased to two folds[16]. Though these mouse models are relatively close to human pathology, optimizing knockdown efficiency for therapeutic effectiveness without completely disrupting the gene is a therapeutic challenge for human CMT1A patients with three copies of the PMP22 gene, and excessive knockdown could cause pathologies similar to the PMP22 deletion (hereditary neuropathy with liability to pressure palsies (HNPP) that cause rare cases of hereditary neuropathy due to the loss of function[21,22]).

A tandem repeat of 1.5 Mb in 17p11.2 causes pathological duplication of the PMP22 gene, resulting in CMT1A[23]. As an underlying mechanism of the low copy number repetitive elements, Pentato and colleagues proposed that these repetitive elements could be induced by rearrangement of CMT1A-REP, two homologous sequences flanking the 1.5-Mb genome unit, during spermatogenesis by unequal crossing-over during meiosis[24]. This was supported by the finding that only paternal duplication occurred in nine sporadic CMT1A patients[25]. Consequently, the mutant CMT1A genome encodes two copies of identical genome sequences of the PMP22 genes.

Hence, one of the most promising treatment approaches for CMT1A is genome editing of the PMP22 gene duplication. Challenges to this gene therapy modality include the need to remove only one copy of the PMP22 gene without disrupting the second copy, and also the efficiencies of infection and genome editing in adult Schwann cells, which produce the myelin sheath of peripheral nerves. YAC transgenic mice[19,20] are highly interesting, while the 560 kb YAC is not sufficiently long and 8 copies of transgene are too much to test our method that need an extra genome copy more than 1.5 Mb. In the present study, we develop an AAV vector that specifically deletes a single copy of the PMP22 gene from duplicate genes present in human iPS cells (iPSCs) derived from a CMT1A patient. This molecular tool overcomes a major roadblock to therapeutic use of this modality, leaving the remaining challenge of in vivo delivery in animal models and then humans, which could be overcome by technical advances expected in the gene therapy field.

## Methods

**sgRNA design and selection.** To target the genomic region of PMP22, a series of sgRNAs were designed by inserting the region between the STR polymorphic genetic markers D17S261 and D17S122 (sgRNA region: 15360587 to 15398063)[26] into Cas-OFFinder (www.rgenome.net/cas-offinder). This region is outside of proximal CMT1A-REP (15575936 to 15587412), distal CMT1A-REP (14179212 to 14190596), and the PMP22 gene (15229777 to 15265357). The following five sequences were selected and subcloned into the pX601 vector (Addgene plasmid #61591) at the BsaI site after addition of a specific protospacer adjacent motif (PAM) sequence for SaCas9:

CMT1AsgRNA1-f: CACCGATATCACTCTCATGACTAGTT
CMT1AsgRNA2-f: CACCGGGCCCAAGGTCTAATTTACAT
CMT1AsgRNA3-f:
CACCGCTGAAAGCATAGTTAGGAAGA
CMT1AsgRNA4-f:
CACCGAACAGGGAAACAAACAGTGGG
CMT1AsgRNA5-f:
CACCGAGAACTGAAAGCATAGTTAGG

The plasmid sequences were validated by the sequencing primer: pX601-sequencing: GAGGTACCTGAGGGCCTATT

For final selection from the five gRNA candidates, a T7E1 mismatch detection assay[27] was used to determine the genome editing efficiency of CRISPR-SaCas9 as indicated by the InDel mutation rate. To calculate the percent cutting efficiency of the CRISPR locus, the intensity of the PCR amplicon band and digested bands were measured with Image J (http://imagej.nih.gov/ij/; National Institutes of Health, USA), and %InDel was calculated by the following formula[28]:

$$\%\text{InDel} = \left(1 - \sqrt{1 - \frac{(a+b)}{(a+b+c)}}\right) \times 100 \qquad (1)$$

**AAV construction.** AAV2 is considered one of the most efficient serotypes for transduction of human Schwann cells[29], so was used it in this study. An AAV Helper-Free System (Agilent Technologies, Santa Clara, California, United States, Catalog #240071) was used to produce AAV particles. The recombinant AAV vectors were produced by transient transfection of HEK293 cells using a vector plasmid; a plasmid for AAV2 Rep and AAV2 Cap expression; and an adenoviral helper plasmid, pHelper. To verify AAV infection efficiency in Schwann cells, the S16 rat Schwann cell line was seeded in an 8-well glass chamber at $4 \times 10^3$ cells/well. Cells were infected with AAV1 virus containing the CAG promoter and EGF (AAV1-CAG-EGF, $2.2 \times 10^{11}$ vg/mL), at MOIs of 5000, 10,000, 20,000, 50,000, 100,000, or 200,000 and incubated for 24, 48, 72, 96, 120, and 144 h.

**iPSC Schwann cell differentiation.** 201B7 iPSCs and HPS0426 PMP22 duplication iPSCs, whose mycoplasma infection was denied by DNA staining (Hoechst 33258) and nested-PCR

methods, were differentiated by modifying the previous method[30]. Briefly, six-well plates were coated with Matrigel GFR diluted 1:20 in DMEM/F12 at RT for 1 h. On day 0, iPSCs were seeded at $2 \times 10^5$ cells/well in Stemfit medium containing Y-27632, a ROCK inhibitor. On the subsequent day, medium was replaced by neural differentiation medium containing 1:1 DMEM/F12: Neurobasal media containing 1:100 N2, 1:50 B27, 0.005% BSA, 2 mM GlutaMAX, 0.11 mM β-Mercaptoethanol, 3 μM CHIR, and 20 μM SB. After 6 days, the cells reached 80–90% confluency and were ready for the first passage into Schwann cell precursor differentiation medium containing equal amounts of DMEM/F12 medium (#11320033 Gibco) and Neurobasal medium (#21103049 Thermo Fisher Scientific) supplement for growth and expression of neuroblastomas (N2, #17502-048 Thermo Fisher Scientific), neuronal cell culture supplement (B27 #17504044 Thermo Fisher Scientific), 0.005% BSA (Bovine Serum Albumin), 2 mM GlutaMAX (#35050061 Thermo Fisher Scientific), 0.11 mM β-Mercaptoethanol (#21985023 Gibco), 3 μM GSK3 inhibitor (CHIR99021, #252917-06-9 Tocris Bioscience), 20 μM TGF-βinhibitor (SB431542, #13031 Cayman), and 50 ng/mL Neuregulin 1 (NRG1, #396-HB-050 R&D Systems). After 18 days, the medium was changed to Schwann cell differentiation medium containing low-glucose DMEM, 1% FBS, 200 ng/mL NRG1, 4 μM Forskolin (#F6886-10MG Sigma), 100 nM RA, and 10 ng/mL PDGF-BB. Media was changed daily for 4 days.

Cells were subsequently incubated in Schwann cell differentiation medium without FK or RA for 2 days. PDGF-BB was then removed, and cells were incubated in low-glucose DMEM (1 mg/mL), 1% FBS, and 200 ng/mL NRG1 (Neuregulin #396-HB-050 R&D Systems). Immunohistochemistry was performed to verify Schwann cells marker expression in differentiated cells.

Cells were infected with AAVs one day after starting iPSC culture (day 1), one day after initiating differentiation into Schwann cell precursors (day 26), and on day 43, after differentiation into Schwann cells. Each AAV was infected at MOI:10,000. DNA from the infected cells was collected on day 50, and the genome-editing efficiency was measured using qPCR.

**CMT1A editing**. AAV infection of CMT1A-iPSCs was performed as follows. Twelve-well plates were coated with Matrigel and seeded with CMT1A iPSCs ($7.5 \times 10^4$ cells/well). Four wells contained cells without virus, four wells cells were infected with AAV encoding control gRNA (AAV2-hSaCas9-gRNAcont, MOI 10,000), and four wells were infected with AAV2-hSaCas9-gRNAedit, MOI 10,000. After 1 week, PCR for AAV-ITR and Nested qPCR were performed to determine the efficiencies for AAV infection genome-editing, respectively.

For AAV-ITR PCR, primers from Riken and Addgene were used. Riken: 5′-GAGTGGCCAACTCCATCACTAGGGGTTC CT-3′. Addgene: fwd ITR primer 5′-GGAACCCCTAGTGATG-GAGTT/ rev ITR primer 5′-CGGCCTCAGTGAGCGA. AAV-ITR PCR was conducted with LA Taq with the following cycles: 94 ℃ 1 min, 35 cycles of 94 ℃ for 1 min, 64 ℃ for 30 s, and 72 ℃ for 3 min, and a final cycle of 72 ℃ for 7 min.

Nested qPCR was performed according to the Human Taqman Copy Number Assay (ThermoFisher). The probe sequences are as follows:

CMT1A Tqm: 6FAM-AAGAAGAATCGTGGGCACACCAC CA-TAMRA

Primers for primary PCR of CMT1A recombination site are:
CMT1A PR1: TGATATTTAAAGATTTCATGTC
CMT1A DF1: GGATTCAGAGACATTAGTGTTCC
Products from the first PCR were digested with *ExoI* (NEB).

Primers designed for secondary PCR of the CMT1A recombination site are below:
CMT1A PR2: CATGTCATTAGACCAAAGAaC
CMT1A DF2: AGAAACATACTAGTTGATATCTTCTaT
RNase P (VIC-TAMRA) was used as an internal control. Both PCR assays were performed with Platinum Taq (TAKARA).

Signal intensity of bands on agar gel was measured using Image J software (http://imagej.nih.gov/ij/. National Institutes of Health, USA).

Quantification of the hybrid region was conducted according to the following formula:

$$Hybrid\_Relative\_Quantification = \frac{2^{-(hybrid-internal\_control)_{AAV\_treated}}}{2^{-(hybrid-internal\_control)_{untreated}}}$$

(2)

**Southern blot analysis**. For probe labeling, genomic DNA prepared from normal iPSCs was used for PCR (KOD Fx Neo) with the following primers: CMT1A_Probe-Fw1: AAGAA-GAATCGTGGGCACAC and CMT1A_Probe-Rv1: AGTG-CAAACCATGATCACCC. The PCR products were purified with the Favorgen kit and labeled with DIG-DNA labeling (Roche DIG labeling). Southern blot was performed using 5 μg iPSC genomic DNA digested with *EcoRI* and *SacI*, electrophoresed in 0.8% agar gel, treated with 0.25 N HCl for depuration and cleaved by alkaline treatment. DNA was transferred from agarose gel to a membrane (Hybond-N, RPN303 N, GE Healthcare, Chicago, IL, USA) at room temperature for 20 h. Prehybridization of the blotted membrane was performed using DIG Easy Hyb, and hybridization was performed by adding DIG-labeled DNA probe (35 ng/mL) at 42 ℃ for 20 h.

**Immunocytochemistry**. Cells were fixed with 4% formaldehyde for 10 min at 25 °C and permeabilized with 0.1% Triton X-100 in PBS for 5 min at 25 °C. After blocking with 10% Fetal Bovine Serum (#10270106, Gibco, MA, USA) for 30 min at 25 °C, cells were incubated with primary antibody for 16 h at 4 °C and with secondary antibodies for 1 h at 25 °C. The antibodies were as follows: rabbit anti-S100B antibody (1:500, ab52642, abcam, Cambridge, UK), goat anti-Sox10 antibody (1:100, sc-17342, Santa Cruz Biotechnology, Dallas, TX, USA), mouse anti-Myelin Basic Protein antibody (1:200, ab62631, abcam, Cambridge, UK), rabbit anti-MAP2 (1:100, sc-32791, Santa Cruz Biotechnology, Dallas, TX, USA), rabbit anti-PMP22 (1:500, ab126769, abcam, Cambridge, UK), Alexa Fluor 647 anti-MAP2 antibody (1:500, ab225315, abcam, Cambridge, UK), Alexa Fluor 488-conjugated anti-rabbit IgG (1:1000, #A21206, Molecular Probes, Eugene, OR, USA), Alexa Fluor 568-conjugated anti-mouse IgG (1:1000, #A10037, Molecular Probes, Eugene, OR, USA) and Cy3-conjugated anti-mouse IgG (1:500, 705-165-003, Jackson Laboratory, Bar Harbor, ME, USA). Images were taken by confocal microscopy (Olympus FV1200IX83, Tokyo, Japan).

**Western blot analysis**. Cells were scraped and collected with PBS. After centrifugation (1500 rpm, 5 min), cell pellets were lysed in sample buffer (25 mM Tris-HCl ph6.5, 5% glycerol, 1% SDS, 1% mercaptoethanol and 0.05% BPB) and heated at 100 ℃ for 5 min. Samples were electrophoresed by SDS-PAGE, and the gels were transferred onto Immobilon-P polyvinylidene difluoride membranes (Millipore, Burlington, MA, USA) using semi-dry transfer, and then blocked with 7.5% milk in TBST (10 mM Tris-HCl pH 8.0, 150 mM NaCl, 0.05% Tween-20). Then the membranes were incubated with mouse anti-PMP22 antibody (1:200, sc-515199, Santa Cruz Biotechnology, Dallas, TX, USA) for 3 h and mouse anti-GAPDH antibody (1:3000, MAB374, sigma-Aldrich, St.

Louis, MO, USA) for 1 h at 25 °C. Then incubated with HRP-linked anti-mouse IgG (1:5000, NA931, GE Healthcare, Buckinghamshire, UK) for 1 h at 25 °C. ECL Select Western Blotting Detection Reagent (RPN2235, GE Healthcare, Chicago, IL, USA) and a luminescent image analyzer (ImageQuant LAS 500, GE Healthcare, Chicago, IL, USA) were used to detect proteins.

**Electron microscopy**. Co-cultures of iPSC-derived neurons and iPSC-derived Schwann cells were pre-fixed with 2.5% glutaraldehyde in 0.1 M phosphate buffer for 2 h at 4 °C, and post-fixed with 1% osmiumtetroxide for 2 h at 4 °C. Following fixation, cells were dehydrated with a graded series of ethanol, and embedded in epon for 48 h at 60 °C, and 24 h at 120 °C. Ultrathin sections (80 nm) were cut with an ultramicrotome (US6, Leica, Wetzlar, Germany) and incubated with uranyl acetate and lead citrate. Sections for Immunoelectron microscopy were incubated in blocking buffer (1% BSA in PBS) for 1 h at 25 °C, and stained with rabbit anti-Crispr-Cas9 antibody (1:25, ab203933, abcam, Cambridge, UK) for 16 h at 4 °C, then incubated with anti-IgG(H + L), Rabbit, Goat-Poly, Gold 10 nm, EM (1:200, EMGAR10, BBI Solutions, Wales, UK), anti-IgG(H + L) for 2 h at 25 °C. Ultrathin sections were observed by electron microscopy (JEM-1400, JEOL, Tokyo, Japan).

**Whole genome sequencing**. For evaluation of off-target effects of genome editing vector in vitro, genome DNA was extracted from control and genome-edited iPS cells according to the protocol described above. For evaluation of off-target effects by intraneural injection of genome editing vector in vivo, $6 \times 10^{10}$ vg of AAV2-hSaCas9-gRNAedit was injected to sciatic nerve of three C57/BL6 mice at 3 months of age, and the injected tissues were dissected after 4 weeks. FavorPrep Tissue Genomic DNA Extraction Mini kit (FATGK001, FAVORGEN, Ping Tung, TAIWAN) was used to extract genome DNA from iPS cells as well as AAV-injected or control non-injected mouse tissues. Whole genome sequencing (WGS) was performed by Illumina NovaSeq 6000 with Truseq DNA Nano in Rhelixa (Tokyo, Japan), at the condition of 150 bp×2 paired-end (PE150), 90 G bases per sample, and 600 M reads per sample (300 M pairs).

The acquired data were analyzed by using HaplotypeCaller function in Genome Analysis Toolkit (GATK, v4.2.3.0), and the results were annotated by using Ensembl Varient Effect Predictor (VEP) (https://asia.ensembl.org/info/docs/tools/vep/index.html)[31] to identify known and unknown SNPs different from the data base (human iPS cells: dbSNP, C57/BL6 mice: Ensembl Variation). Original SNPs found in the non-infected controls were further excluded to identify candidate SNPs for de novo mutations by genome editing. Genome regions duplicated in human CMT1A iPS cells were identified from the increase of reads in WGS. To visualize the increase of reads, IGVTools (Broad Institute, v2.5.3) was used to calculate read depth from WGS data.

**TIDE for evaluation of genome editing efficiency**. TOPO subcloning and Sanger sequencing were performed following the protocol (https://www.thermofisher.com/jp/ja/home/life-science/genome-editing/genome-editing-learning-center/genome-editing-resource-library/genome-editing-application-notes/sanger-sequencing-facilitate-crispr-talen-mediated-genome-editing-workflows.html). Genomic DNA prepared from CMT1A-iPSCs 2 weeks following AAV2-hSaCas9-gRNAedit infection was amplified by junction 4F (TGGATGGTGGTAGGTATCATTCA) and Junction 4R (TGGGGCACATGAGATATTTTGG) primers. The amplified DNA fragments were subcloned into pCR®4-Blunt TOPO®, and 1,000 clones were sequenced by Sanger method at

Eurofins Genomics (Tokyo, Japan). The sequence data were analyzed by BLAST at the supercomputer SHIROKANE (The Institute of Medical Science, The University of Tokyo).

**Karyotype analysis**. A G-band analysis was performed to determine the karyotype of the iPSC line. Twenty metaphase plates were analyzed.

**Statistics and reproducibility**. Statistical analyses for biological experiments were performed using Graphpad Prism 8. Biological data following a normal distribution are presented as the mean ± SEM, with Tukey's HSD test for multiple group comparisons or with Welch's t-test for two group comparisons. The distribution of observed data was depicted with box plots, with the data also plotted as dots. Box plots show the medians, quartiles, and whiskers, which represent data outside the 25th–75th percentile range. Data not following a normal distribution are examined by Wilcoxon's rank sum test with post-hoc Bonferroni correction. To obtain each data, we performed biologically independent experiments. The number of samples was indicated in each figure and figure legends.

**Ethics declarations**. This study was performed in strict accordance with the recommendations of the Guide for the Care and Use of Laboratory Animals of the Japanese Government and the National Institutes of Health. Animal experiments were performed in accordance with the ARRIVE guidelines. All experiments were approved by The Committees for Gene Recombination Experiments, Ethics and Animal Experiments of the Tokyo Medical and Dental University (G2018-082C3, O2017-008, and A2021-211A). Normal and *CMT1A* iPSC (201B7 and *PMP22* duplication iPSCs) were derived from Cell Bank of RINEK BRC (HPS0063 and HPS0426), and researches with these iPSCs were approved by The Ethics Committee of Tokyo Medical and Dental University (O2017-008). Informed consent was obtained from all participants for generation of iPSCs.

**Reporting summary**. Further information on research design is available in the Nature Portfolio Reporting Summary linked to this article.

## Results

**Development of genome-editing AAV vectors targeting PMP22 gene duplication**. A tandem repeat of 1.5 Mb in 17p11.2 including two *PMP22* gene copies causes CMT1A[23] (Fig. 1a). To cleave out a *PMP22* copy by genome editing, a series of sgRNAs were designed to target sequences in the 1.5-Mb genome unit other than the *PMP22* gene or CMT1A-REP using Cas-OFFinder (www.rgenome.net/cas-offinder)[32] software, and among to twenty sequences five candidate sequences were further selected based on the identity at two positions (Supplementary Fig. 1a). The five candidate sequences were added with a specific protospacer adjacent motif (PAM) sequence for SaCas9 and subcloned into the BsaI site of the pX601 vector (Addgene plasmid #61591) (Supplementary Fig. 1a). Plasmids were transfected into normal iPSCs (201B7 from RIKEN cell bank) to determine if the 5' genome sequence of *PMP22* gene could be cleaved by genome editing (Supplementary Fig. 1b, c). SaCas9-mediated cleavage efficiency (percentage InDel) was calculated using a T7E1 assay based on the fraction of cleaved DNA, as determined by the integrated intensity of the gel bands (Supplementary Fig. 1d). The relative amount of cleaved DNA fragments detectable on the gel was considered an indicator of mutation frequency within the cell population. SaCas9-mediated cleavage efficiency (percentage InDel), which was calculated based on the fraction of cleaved

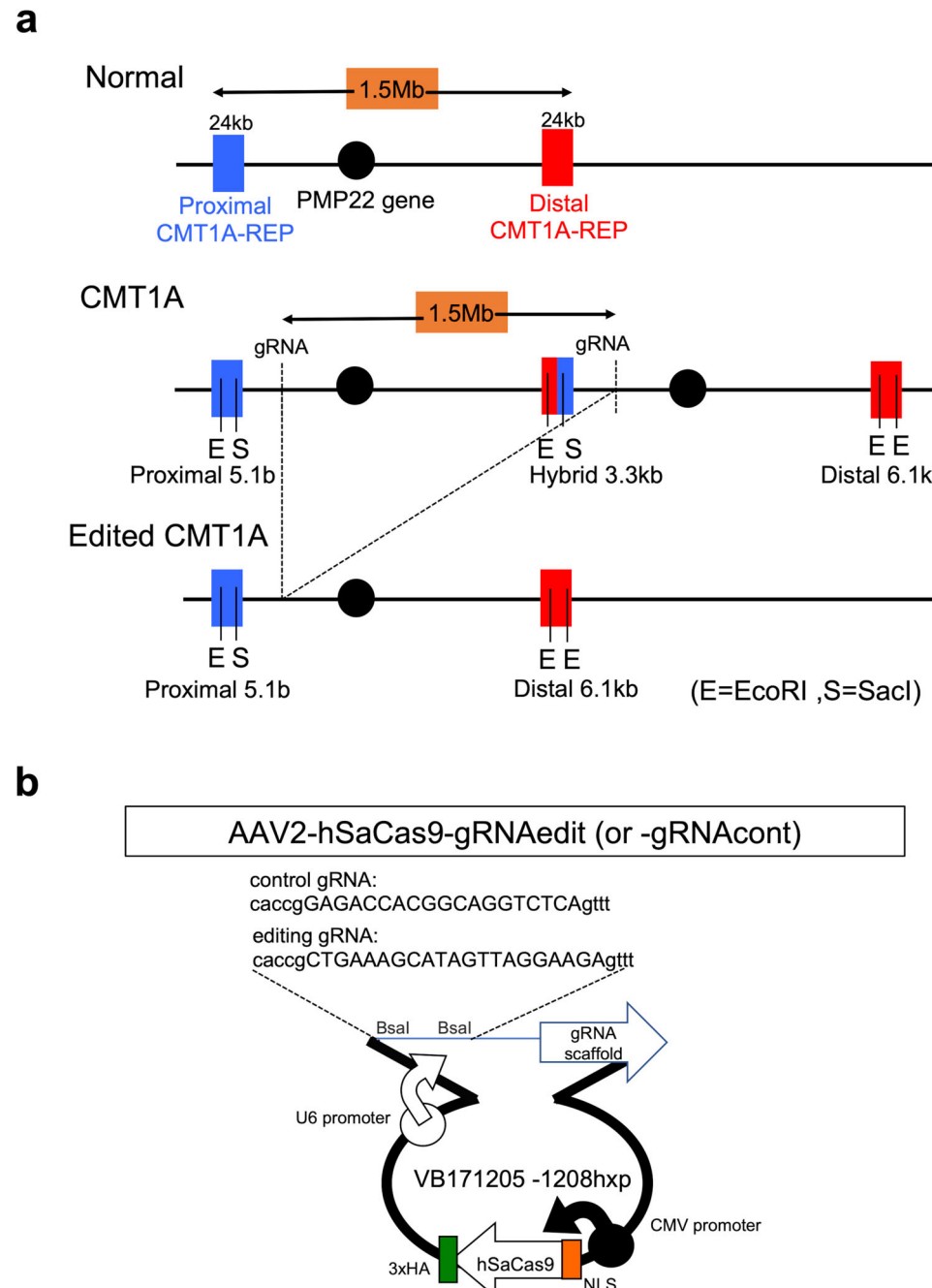

**Fig. 1 AAV-mediated genome editing strategy. a** Human genome structures surrounding the *PMP22* gene of normal, CMT1A, and post-editing CMT1A cells with target sites for genome editing. **b** Map of the genome editing VB171205 plasmid used to remove additional *PMP22* gene copies from CMT1A cells, from which AAV2 vector was generated in HEK293 cells.

DNA as determined by the integrated intensity of the gel bands, was highest with pX601-sgRNA3 (20.18%), followed by pX601-sgRNA5 (18.63%), and pX601-gRNA2 (12.36%). Other vectors (pX601-sgRNA1 and pX601-sgRNA4) did not produce cleavage in 201B7 iPSCs. In 293 T cells, pX601-sgRNA5 was the most efficient in DNA cleavage (33.50%) followed by pX601-sgRNA3 (33.21%), pX601-sgRNA1 (27.55%), and pX601-sgRNA2 (16.73%). pX601-sgRNA4 did not have detectable activity in 293T cells. Based these findings, CMT1A-sgRNA3 was selected to generate the AAV2 vector (AAV2-hSaCas9-gRNAedit) for subsequent experiments (Fig. 1b, Supplementary Fig. 1e).

**Correction of PMP22 gene duplication in CMT1A-iPS cells.**
We next determined whether our AAV vector could correct duplication of the human *PMP22* gene using human CMT1A iPS cells (Figs. 2, 3). Human iPSC or iPSC-differentiated cells are considered to be the optimal model for genetic events underlying CMT1A, because no humanized disease genome model carrying duplicated human PMP22 genome together with long flanking regions of human genome around *PMP22* gene is currently available and because Human iPSC or iPSC-differentiated cells are considered a simple system to test genetic events underlying CMT1A. Moreover, this was the most direct means to test the ability of our vector to edit the human genome, rather than that

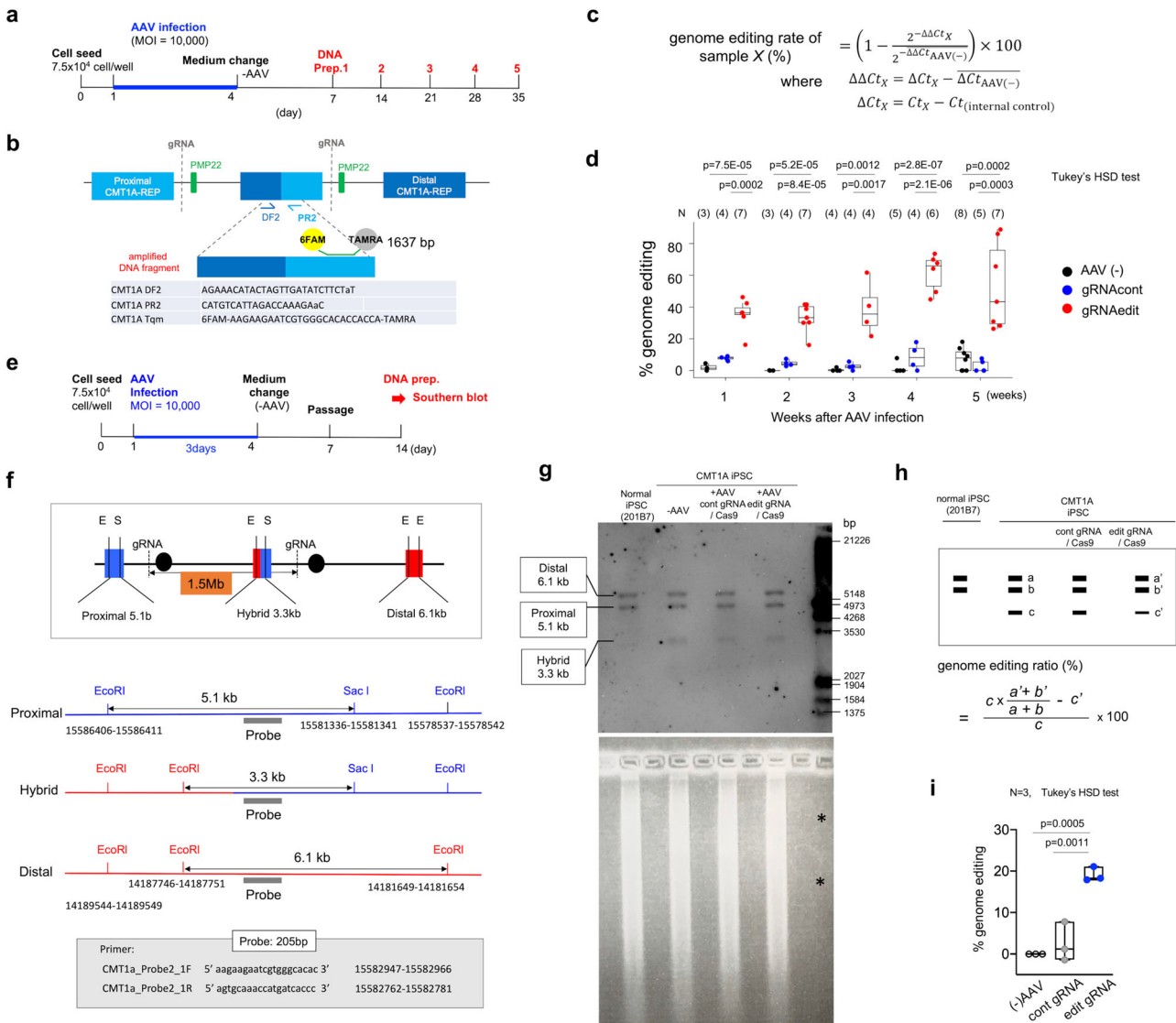

**Fig. 2 Verification of genome editing in human CMT1A iPS cells. a** Protocol for genome editing verification in human CMT1A iPS cells by quantitative PCR. **b** Method and secondary primers for nested PCR used to detect non-edited hybrid CMT1A-REP. **c** Formula to calculate percent genome editing. Ctx: Ct value of sample X, Ct (internal control): Ct value of internal control in qPCR for sample X. **d** Percent genome editing of iPSCs after 1, 2, 3, 4, or 5 weeks of infection with AAV2 genome-editing vector. Number of samples (AAV-, gRNAcont, gRNAedit) were (3, 4, 7) at 1 week, (3, 4, 7) at 2 weeks, (4, 4, 4) at 3 weeks, (5, 4, 6) at 4 weeks and (8, 5, 7) at 5 weeks. *P*-values in Tukey's HSD test are shown in the graph. **e** Protocol for verification of genome editing in human CMT1A iPS cells with Southern blot analysis. **f** Left scheme shows genome structure and proximal, hybrid and distal CMT1A-REP. EcoRI-SacI digestion produced 5.1-kb, 6.1-kb and 3.3-kb bands from three REP sequences. **g** Upper image showed Southern blot result detecting the three bands. Lower image showed agar gel stained by ethidium bromide, revealing equal amounts of genomic DNAs were loaded. Asterisks indicate that 1/50 amount of Digoxigenin-labeled DNA molecular weight marker III was not visible by ethidium bromide staining. **h** Strategy to calculate percent genome editing by Southern blot after correction for blot and hybridization efficiencies. **i** Quantitative analysis of percent genome editing in non-infected, AAV2-hSaCas9-gRNAcont-infected, and AAV2-hSaCas9-gRNAedit-infected iPS cells. *N* = 3 each. *P*-values in Tukey's HSD test are shown. The box plot shows median, 25–75th percentile, and whiskers representing data outside the 25–75th percentile range.

of an animal model. Therefore, an iPSC line (HPS0426) generated from a patient (40–49 years old, female) with duplication of the *PMP22* gene in CiRA at Kyoto University (CiRA00139) was obtained from the RIKEN cell bank. Karyotype of the CMT1A-iPSCs (from here CiRA00139 is designated as CMT1A-iPSCs) was normal (Supplementary Fig. 2a). Duplicated region was mapped to Chr17: 14,173,927–15,588,542 by whole genome sequencing (Supplementary Fig. 2b), indicating that the CiRA00139 cell line possessed an ordinary mutation as CMT1A. Duplication of the *PMP22* gene was heterozygous judging from the patient family history and from 1.5 folds increase of reads in whole genome sequencing (Supplementary Fig. 2b).

Subsequently, CMT1A-iPSCs were infected with AAV2-hSaCas9-gRNAedit or AAV2-hSaCas9-gRNAcont (Fig. 2a). Non-edited distal-proximal fusion CMT1A-REP, which is specific for the disease allele, was detected by quantitative PCR (Fig. 2b), and the efficiency of genome editing was expressed as percent genome editing (Fig. 2c). CMT1A-iPSCs were infected with AAV vectors for 3 days, incubated in normal media for a subsequent 3 days and passaged. DNA from CMT1A-iPSCs was sampled weekly following AAV infection (Fig. 2a). This approach reached nearly 40% of genome editing efficiency, while the values became unstable after 4 weeks of infection, as iPSC viability declined (Fig. 2d, Supplementary Fig. 3).

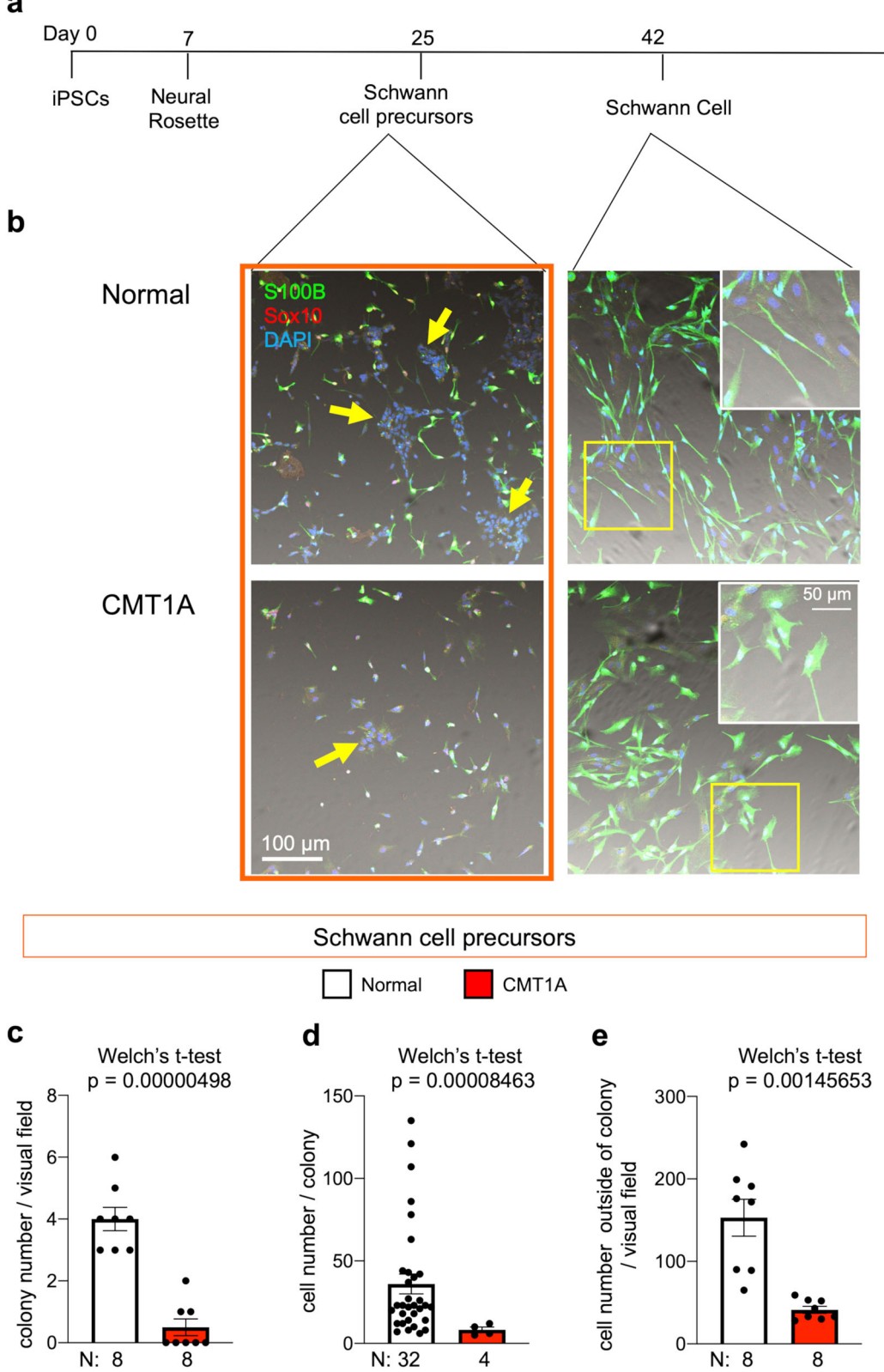

**Fig. 3 Schwann cell differentiation from normal or CMT1A iPSCs. a** Protocol for Schwann cell differentiation from iPS cells. **b** Representative images of cultures at Schwann cell precursor (Day 25) and Schwann cell (Day 42) stages. **c–e** Quantitative analysis of colony numbers (**c**), colony cell proliferation (**d**), and Schwann cell precursor numbers outside of colonies (**e**). Number of samples were 8 each in colony number / visual field, 32 (normal) and 8 (CMT1A) in cell number / colony, and 8 each in cell number outside of colony / visual field. Welch's $t$-test was performed in each graph. Bar plots indicate the mean $+/-$ SE. Raw data points are shown in dots.

To confirm the genome editing by the AAV2-hSaCas9-gRNAedit vector, we performed Southern blot analysis of genomic DNA prepared from CMT1A-iPSCs 2 weeks following AAV infection (Fig. 2e). EcoRI-SacI co-digestion of genomic DNA generated from proximal, hybrid, or distal CMT1A-REP-infected cells generated different DNA fragment lengths (Fig. 2f, g). The signal intensity of a 3.3-kb DNA fragment (band c) after correction with the signal intensities of a 6.1-kb fragment (band a) and 5.1-kb fragment (band b) was used as a quantitative indicator of non-edited genome (Fig. 2g, h). By subtracting the 3.3-kb DNA fragment intensity of CMT1A-iPSCs after AAV2-hSaCas9-gRNAedit-infection (c') from the corrected intensity of band c, the percentage genome editing can be calculated (Fig. 2h). In this Southern blot-based assay, nearly 20% of genome editing following AAV2-hSaCas9-gRNAedit infection was confirmed (Fig. 2i). Southern blot is influenced by many factors such as DNA digestion efficiency, blotting efficiency and hybridization homogeneity. Therefore, the discrepancy of values between the two methods would be permissible, and the percentage genome editing was considered roughly above 20%.

However, an exactly precise quantification of editing efficiency based on qPCR or Southern blotting is not possible. Detection of a small increase is challenging in qPCR, and the intensity of bands in Southern blotting might not accurately correlate with the count of edited genomes. Therefore, further to examine the efficiency of genome editing, we employed the third method TIDE (Tracking of Indels by Decomposition)[33], TOPO subcloning and Sanger sequencing with genomic DNA prepared from CMT1A-iPSCs 2 weeks following AAV2-hSaCas9-gRNAedit infection (Supplementary Fig. 4a). Pilot study of 4 plasmids revealed that two plasmids from AAV-infected CMT1A-iPSCs contained amplified genomes from different chromosomes. Meanwhile the other two plasmids contained the exact target region of the *PMP22* gene, both of which indicated occurrence of genome editing (Supplementary Fig. 4b). Therefore, we extend the analysis to 1,268 plasmids (Supplementary Fig. 4c). The result revealed a genome editing efficiency of 16.77% that was lower than the values obtained by the former two methods. However, mutations could not be counted by TIDE when the Cas9-mediated cleavage was reunited without generating point mutations during deletion of 1.5 Mb genome region. Therefore, the efficiencies of genome editing evaluated by three methods were considered consistent in a large sense.

**Abnormal proliferation and differentiation of Schwann cell precursors from CMT1A-iPS cells.** A previously published protocol[30] was used to differentiate normal iPSCs and CMT1A iPSCs to Schwann cell precursors (SCP) and subsequently into Schwann cells (Supplementary Fig. 5). Compared with normal iPSCs, the CMT1A iPSCs exhibited delayed proliferation of Sox10-positive, SCPs and the morphology of S100B-positive Schwann cells was abnormal, as evidenced by the larger gemistocytic shapes (Fig. 3a, b). Quantitative analyses confirmed these Schwann cell characteristics during differentiation of human CMT1A iPSCs, including decreased proliferation from iPSCs to SCPs (cell counting) and delayed differentiation (decreased expression of Schwann cell marker S100B and SCP marker Sox10) (Fig. 3c–e). Meanwhile, normal iPSCs differentiated into Schwann cells with thin and fibrillary shapes. These findings supported the notion that CMT1A pathology begins at early developmental stages, as suggested by previous reports[34,35].

**Abnormal myelination and cell death in co-cultures of CMT1A-iPSC-derived Schwann cells and normal iPSC-derived pan-neurons.** Subsequently, we examined how abnormal Schwann cell differentiation of CMT1A-iPSCs affected the functional outcome of neurite myelination. iPSC-derived neurons and iPSC-derived Schwann cells were co-cultured according to our original protocol developed in this study (Fig. 4a), in which we combined the protocol to differentiate iPSCs to pan-neurons[36] and the protocol to co-culture rat dorsal root ganglion (DRG) neurons and human iPSC-derived Schwann cells[30]. Neurons were differentiated only from normal iPSCs, while Schwann cells were differentiated from both normal and CMT1A iPSCs to determine the myelination function of disease-linked Schwann cells (Fig. 4a).

Co-immunostaining of myelin basic protein (MBP), a marker of mature myelin, and microtubule-associated protein 2 (MAP2), a neuronal marker, revealed myelinated or hypomyelinated neurites (yellow and green arrows, Fig. 4b, upper panels) in co-cultures of normal iPSC-derived neurons and normal iPSC-derived Schwann cells (Fig. 4b, upper panels). To compare the extent of myelination, the percentage of MBP-MAP2 merged lengths relative to total neurite length was quantified in images with a constant area (Fig. 4b, lower panels). The percentage of MBP-MAP2 merged neurite length was significantly different between co-cultures of normal iPSC-derived Schwann cells and co-cultures of CMT1A-iPSC-derived Schwann cells (Fig. 4b, graph).

To investigate the detailed morphology of co-cultures, including the myelination state at co-culture endpoint, transmission electron microscopy (EM) was used (Fig. 4c). In CMT1A-iPSC-derived Schwann cell co-culture, the number of dead cells with apoptotic indicators such as chromatin condensation and apoptotic bodies was statistically increased (Fig. 4c). Meanwhile, neurons with larger nuclei remained alive, suggesting that the dying cells were primarily CMT1A-iPSC-derived Schwann cells (Fig. 4c).

**Rescue of abnormal co-culture phenotypes by AAV-mediated genome editing.** Finally, we determined if genome editing using AAV2-hSaCas9-gRNAedit to normalize duplication of the 1.5-Mb genome fragment would ameliorate the aberrant phenotypes of co-cultures of normal iPSC-derived neurons and CMT1A-iPSC-derived Schwann cells (Fig. 5). Prior to analysis, we examined the genome-editing of this platform by infection at different time points during differentiation, and found that the level of genome editing was highest in Schwann cell precursors (mean 75%), but the level of Schwann cells (42%) and that of iPSC (50%) were still comparable (Supplementary Fig. 6a). Therefore, we determined to infect cells after Schwann cell differentiation that is most applicable for human therapy in the future. At the same time, we observed that CMT1A-iPSC differentiation into S100B-positive Schwann cells was less efficient using the same protocol, which might have implications for therapeutic applications in the future (Supplementary Fig. 6b). Infection of post-differentiation CMT1A-iPSC-derived Schwann cells with AAV2-hSaCas9-gRNAedit but not AAV2-hSaCas9-gRNAcont (Fig. 5a) decreased the cell death ratio in neuron-Schwann cell co-cultures (Fig. 5b).

In addition, MBP and MAP2 co-immunostaining revealed that the gene-editing vector increased the percentage of MBP-MAP2 merged neurite length, indicating improved myelination ability (Fig. 6a). Moreover, EM revealed immature myelination of neuronal process (neurites) in co-culture (Fig. 6b). As shown in the lower cartoon of Fig. 7a, multiple layers of premature myelin sheath surround neuronal processes (Fig. 6b). This complex of neurite and immature myelin was decreased (frequency per area) in co-cultures of normal iPSC-derived neurons and CMT1A-iPSC-derived Schwann cells (Fig. 6c), but was restored in

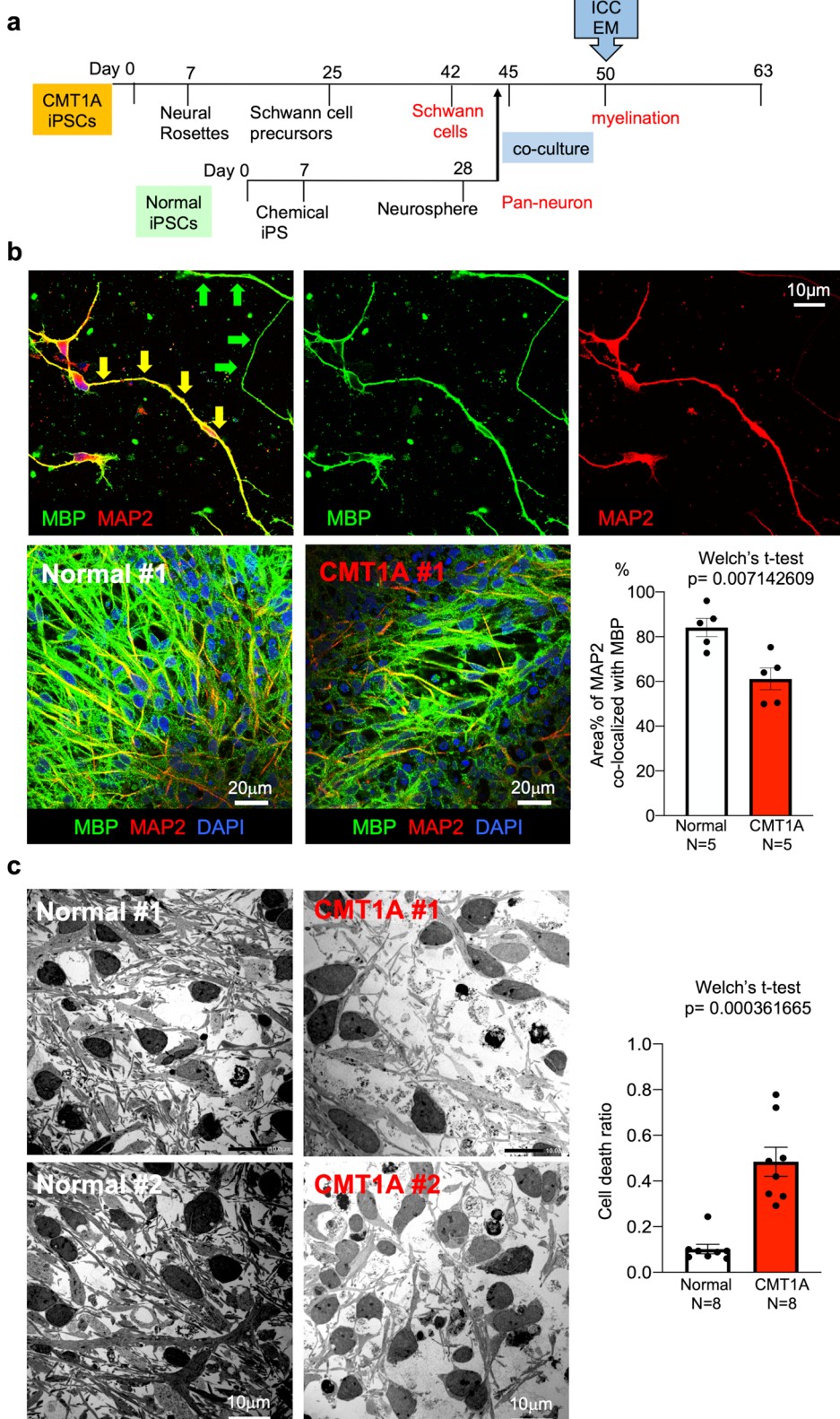

**Fig. 4 Neuron-Schwann cell co-culture of normal or CMT1A-iPS cells. a** Protocol for myelination measurement with neuron-Schwann cell co-culture of normal iPSC-derived neurons and normal or CMT1A-iPSC-derived Schwann cells. **b** MBP and MAP2 co-staining revealed myelinated (yellow arrow) and un-myelinated (white arrow) neurites. At high cell densities (4000 neurons + 40,000 Schwann cells/well in a six-well plate), percent myelination (myelinated fiber area/MAP2-positive fiber area *100) was calculated in co-cultures of normal neurons with normal Schwann cells and normal neurons with CMT1A Schwann cells. $N = 5$ each. Welch's t-test was performed. **c** Electron microscopy revealed an increased ratio of dead cells relative to total cells in co-culture of normal neurons with CMT1A Schwann cells. The right graph shows quantitative analysis of cell death in eight independent areas. $N = 8$ each. Welch's t-test was performed. Bar plots indicate the mean $+/-$ SE. Raw data points are shown in dots.

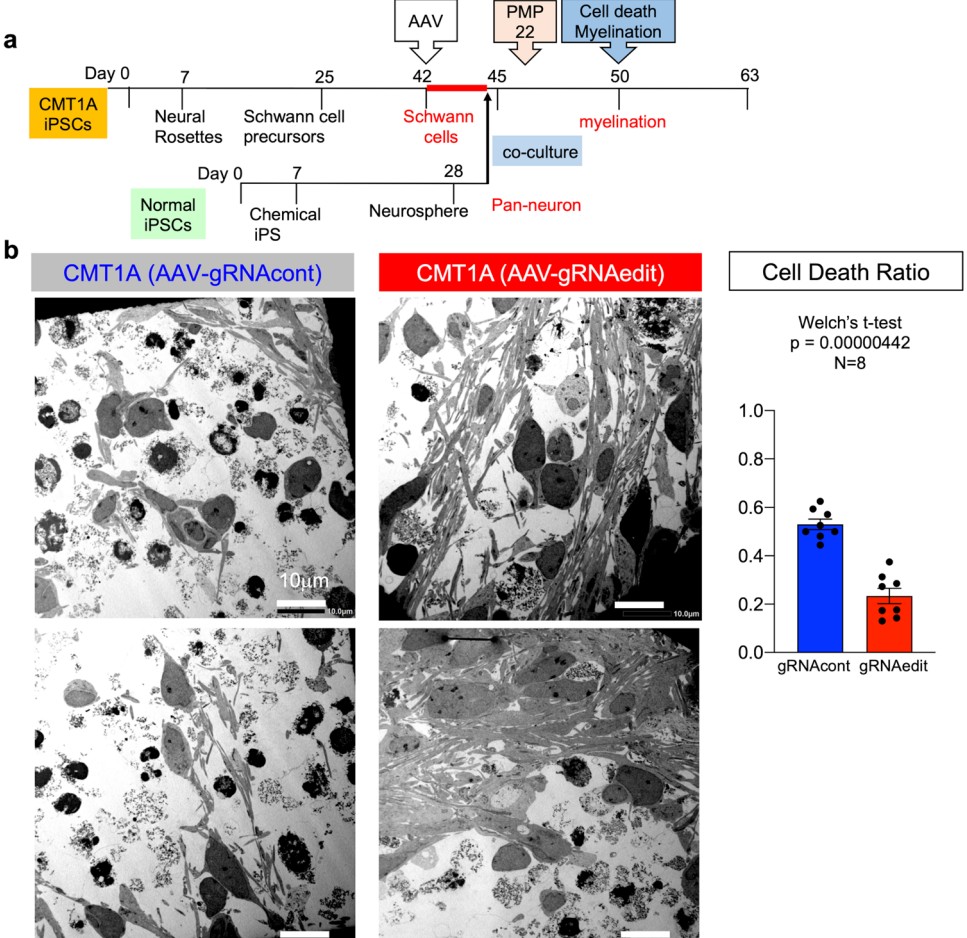

**Fig. 5 Rescue of cell death in CMT1A-iPSC-derived Schwann cells by the genome-editing AAV vector. a** Rescue of co-culture phenotypes of normal iPSC-derived neurons and CMT1A-iPSC-derived Schwann cells by infection with the genome-editing AAV vector in the Schwann cell stage. **b** Two representative EM images of co-culture of normal iPSC-derived neurons and CMT1A-iPSC-derived Schwann cells after infection with AAV2-hSaCas9-gRNAcont (AAV-gRNAcont) or AAV2-hSaCas9-gRNAedit (AAV-gRNAedit). Right graph shows quantitative analysis of cell death ratio (dead cell number/total cell number) calculated from eight EM images. N = 8 each. Welch's *t*-test was performed. Bar plots indicate the mean +/− SE. Raw data points are shown in dots.

co-cultures of normal iPSC-derived neurons and CMT1A-iPSC-derived Schwann cells infected with AAV2-hSaCas9-gRNAedit (Fig. 6c, d).

Moreover, we confirmed that the AAV genome-editing vectors successfully infected Schwann cells used in the above experiments using immunoelectron microscopy with anti-Crispr-Cas9 antibody (Supplementary Fig. 7).

**Effect of genome editing on chromosome, genome sequence, and PMP22 expression**. Finally, we examined possible side effects of our genome-editing method. First, karyotype analysis revealed that the chromosomes were normal after AAV-mediated genome-editing (Supplementary Fig. 8a). Second, whole genome sequencing of genome DNA prepared from $1 \times 10^{6}$ iPSC cells revealed that genome structures were not changed at the chromosome or copy number variation (CNV) level after AAV-mediated genome-editing (Supplementary Fig. 8b). Third, whole genome sequencing was also used to evaluate off-target effects after AAV-mediated genome-editing (Supplementary Fig. 8c, d). We found 66 bases of possible off-target mutations among 3,099,734,149 bp of human whole genome in CMT1A-iPSCs after in vitro AAV2-hSaCas9-gRNAedit-infection (Supplementary Fig. 8c), and 797 bases of possible off-target mutations among 2,728,206,152 base pairs of

mouse whole genome in mouse tissues after AAV2-hSaCas9-gRNAedit-injection to sciatic nerve (Supplementary Fig. 8d). Among 66 possible point mutations, 65 were mapped to introns/non-coding regions, and only 1 was mapped to a protein coding exon of *FCGBP* gene but the coded amino acid was not changed. Final percentages of possible off-target effects on coding and regulatory regions were $3.23 \times 10^{-8}$% in human iPSC and $4.58 \times 10^{-6}$% in mouse tissues (Supplementary Fig. 8c, d). Given that they were all point mutations and no sequential or cluster mutations were found, we could not exclude the possibility that they were de novo SNPs unrelated to genome editing. However, even though the ratios of possible off-target mutations were very low in both cases, off-target effect is a critical issue in clinical application of genome editing technique[37–39] and should be further reduced by development of genome editing techniques[39–41].

We also examined *PMP22* mRNA and protein levels in iPSC-derived Schwann cells before AAV-infection (Day 42) and after co-culture with normal iPSC-derived neurons (Day 47) (Fig. 7a). Immunohistochemistry revealed that PMP22 protein levels were increased in CMT1A-iPSC-derived Schwann cells at 42 days (Fig. 7b). Spheroidal shapes of MAP2-positive/PMP22-positive cells were observed in co-culture of normal iPSC-derived Schwann cells, while they were decreased in co-culture of CMT1A-iPSC-derived Schwann cells (Fig. 7b). Instead, abnormally shrunk shapes of

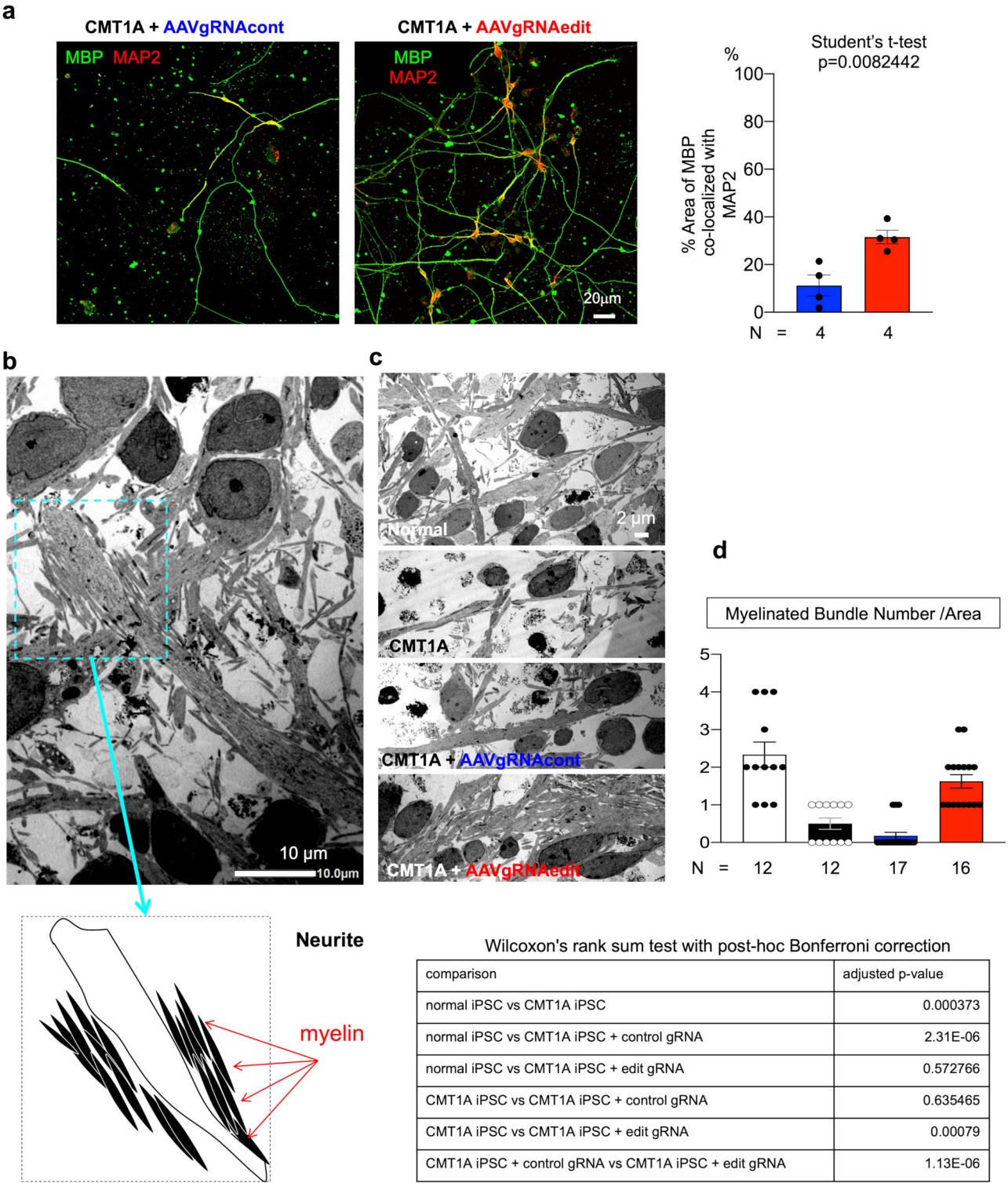

**Fig. 6 Rescue of CMT1A-iPSC-derived Schwann cell myelination function by the genome-editing AAV vector. a** Immunohistochemistry of co-cultures of normal iPSC-derived neurons and CMT1A iPSC-derived Schwann cells after infection with AAV2-hSaCas9-gRNAcont (AAV-gRNAcont) or AAV2-hSaCas9-gRNAedit (AAV-gRNAedit). Quantitative analysis revealed that infection with AAV2-hSaCas9-gRNAedit improved CMT1A Schwann cell myelination function. $N = 4$ each. Student's $t$-test was performed. **b** Representative image including myelinated fibers. In our observation up to Day 50 after seeding iPSCs, while compact myelin sheaths were not observed, loose myelination surrounding neurites was observed. **c** Representative images of co-culture of normal neurons and normal Schwann cells, co-culture of normal neurons and CMT1A Schwann cells, co-culture of normal neurons and CMT1A Schwann cells infected with AAV2-hSaCas9-gRNAcont, and co-culture of normal neurons and CMT1A Schwann cells infected with AAV2-hSaCas9-gRNAedit. **d** Quantitative analysis of myelinated fiber bundles in the four groups shown in c. The lower table shows statistical analysis results. Number of samples were 12 (normal), 12 (CMT1A), 17 (CMT1A + control gRNA), and 16 (CMT1A + edit gRNA). Wilcoxon's rank sum test with post-hoc Bonferroni correction was performed. Bar plots indicate the mean +/− SE. Raw data points are shown in dots.

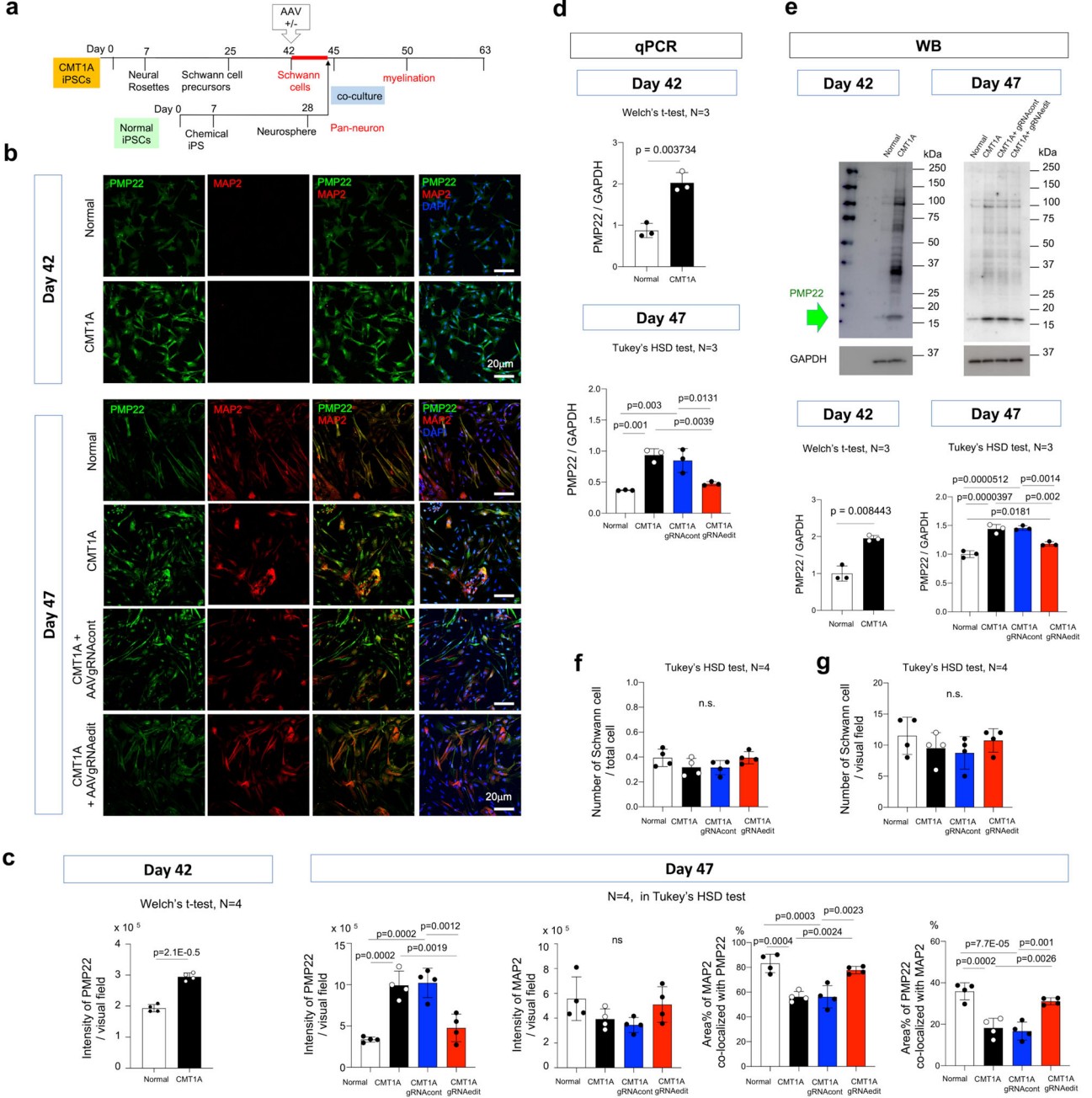

**Fig. 7 Effect of genome editing on expression levels of PMP22. a** Sampling protocol for evaluation of the protein and mRNA expression levels of PMP22. **b** Immunocytochemistry to evaluate the protein level of PMP22 in single-culture of CMT1A-iPSC-derived Schwann cells on Day 42 and colocalization of PMP22 and MAP2 signals in co-culture of iPSC-derived neurons and CMT1A-iPSC-derived Schwann cells on Day 47. PMP22 signals were increased in CMT1A-iPSC-derived Schwann cells than in normal iPSC-derived Schwann cells. The similar increase of PMP22 signals were observed in co-culture, while it was corrected and merged signals of MAP2 and PMP22 were increased by infection of AAV-gRNAedit. **c** Quantitative analysis of PMP22 signal intensities, MAP2 signal intensities, area% of MAP2 co-localized with PMP22, and % Area of PMP22 co-localized with MAP2 in four groups. Welch's *t*-test was used for Day 42 and Tukey's HSD test was used for Day 47. *N* = 4 each. *P*-values are shown in graphs. **d** Quantitative PCR analysis of *PMP22* mRNA in four groups of co-culture. Welch's *t*-test was used for Day 42 and Tukey's HSD test was used for Day 47. *N* = 3 each. *P*-values are shown in graphs. **e** Western blot analysis of PMP22 protein in four groups of co-culture. Welch's *t*-test was used for Day 42 and Tukey's HSD test was used for Day 47. *N* = 3 each. *P*-values are shown in graphs. **f, g** Quantitative analysis of PMP22-positive Schwann cell numbers per total cells (**f**) and per visual fields (**g**) in four groups by Tukey's HSD test. Bar plots indicate the mean +/− SE. Raw data points are shown in dots.

MAP2/PMP22-double positive cells were detected in co-culture of CMT1A-iPSC-derived Schwann cells (Fig. 7b). AAV2-hSaCas9-gRNAedit but not AAV2-hSaCas9-gRNAcont increased spheroidal cells (Fig. 7b). Consistently, quantitative analyses revealed the recoveries of PMP22 signals and of merged area of PMP22 and MAP2 (Fig. 7c). Quantitative PCR and western blot analysis revealed

that both *PMP22* mRNA and protein levels were increased in CMT1A-iPSC-derived Schwann cells, while AAV-mediated genome editing almost normalized the PMP22 levels (Fig. 7d, e). The recovery of the PMP22 levels was not due to the decrease of Schwann cell numbers after infection of AAV2-hSaCas9-gRNAedit (Fig. 7f, g).

## Discussion

PMP22 is a compact myelin protein as well as myelin protein zero (P0), myelin protein 2 (P2) and myelin basic protein (MBP)[42–45], and these molecules respectively form different layers of myelin structural unit[46]. *PMP22* gene dose is related to myelination phenotype, as aberrant duplication of the *PMP22* gene causes hypomyelination and subsequent axonal degeneration, while *PMP22* deletion causes pressure-sensitive neuropathy sometimes with hypermyelination[21]. CMT1A is the most common CMT subtype, accounting for more than 50% of familial peripheral neuropathies[1]. PMP22 alters membrane architecture, with effects dependent on the ratio of lipids to PMP22 protein[47]. However, the specific molecular mechanism by which *PMP22* gene duplication causes hypomyelination and axonal loss is incompletely understood. Some point mutations in the *PMP22* gene disrupt intracellular PMP22 protein transport[48,49]. However, whether a 1.5-fold increase of PMP22 protein levels in human CMT1A causes protein misfolding or induces endoplasmic reticulum associated degradation (ERAD) leading to the decrease of PMP22 protein is unknown.

Impaired proliferation and apoptotic cell death of differentiating Schwann cells is a potential characteristic of CMT1A pathology[34], which is supported by previous studies including retrovirus vector-mediated *PMP22* gene transfer in cultured Schwann cells[50], Schwann cells isolated from nerve biopsies of CMT1A patients[51], and PMP22 overexpression in Schwann cells and NIH-3T3 cells[52,53]. The findings of the present study, which evaluated the phenotypes of human CMT1A-iPSC-derived Schwann cells in monoculture and in co-culture with normal human iPSC-derived pan-neurons demonstrated suppressed proliferation and apoptotic cell death during differentiation to mature Schwann cells. This further suggests that these processes are characteristic of Schwann cells derived from CMT1A patients.

No therapeutics that directly target CMT1A have proven clinical effects on CMT1A. As summarized in recent reviews[9,10], various therapeutics currently developed include indirect methods to increase viability of Schwann cells and/or neurons and direct approaches to decrease PMP22 protein expression. Preclinical studies in rodent models suggested that soluble neuregulin-1[54,55], AAV1-mediated delivery of neurotrophin-3 (NT3)[56], P2X7 receptor inhibition[57], shRNA[15], AAV-mediated expression of miRNA[16], and antisense oligonucleotides targeting PMP22[7] could improve CMT1A pathology. However, these findings have not been confirmed in clinical trials. Meanwhile, clinical trials including ascorbic acid[58,59] (NCT00484510), a myostatin inhibitor[60] and progesterone receptor antagonist (NCT02600286) have failed to identify clinical benefits for CMT1A patients.

The findings of the present study developed a gene therapy technology for removal of the aberrant *PMP22* gene copy that causes CMT1A by genome editing using iPSC-derived Schwann cells from CMT1A patients. The cleavage sites for this genome-editing technology are also located in non-coding and non-functional genome regions, such that a single cleavage of the genome is non-toxic. Deletion of a *PMP22* gene copy by the editing vector occurs only when two sites are cleaved in an abnormal allele with duplicate *PMP22* genes, supporting the safety of this approach. The genome-editing efficiency of the vector is ~20–40% in CMT1A-iPSC-derived Schwann cells. This could be a barrier to clinical application of this tool, although further optimization could improve efficiency. However, an overly effective vector would be also inappropriate for early clinical trials, as this could potentially cause unexpected side effects.

The infection method, i.e., intravenous or intraneural injection, should be considered. A previous study revealed that intravenous injection of PMP22-targeting siRNA conjugated to squalenoyl nanoparticle was short lasting, and needed multiple injections[14].

Another previous study demonstrated that directly injecting an AAV2 vector into peripheral nerves of adult mice yielded a high infection efficiency of mature Schwann cells[15], suggesting this could be an optimal delivery approach in adulthood.

The timing of injection into human CMT1A patients must also be considered. The genome-editing efficiency of the vector developed in the present study was not largely changed by injection of the vector into early iPSCs or to mature Schwann cells. Considering the developmental disease pathology that occurs during Schwann cell differentiation and the relatively high genome-editing efficiency of our vector at SCPs, intervention at an earlier age could increase clinical efficacy. However, ethical issues should be carefully considered for this protocol.

Because no animal model with *PMP22* gene duplication is available by which we can examine the genome editing of our vector, it may be necessary to proceed to human clinical trials for the therapeutic effect after carefully excluding toxicity of our vector in animals and checking the off-target effect of expressed gRNA, even though the frequency of off-target mutations was very low both in vitro and in vivo in our study.

## Data availability

Source data underlying the figures are provided in the Supplementary Data 1 file. Digital uncropped images of gels and blots are shown in Supplementary Fig. 9. Raw sequencing data generated in this study are deposited in the NCBI SRA database (accession code PRJNA1007436), and raw image data are deposited in Image Data Resource (accession code: 10.6084/m9.figshare.24228670). Other data obtained from the current study is available from the corresponding author upon reasonable request.

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

## Acknowledgements
This work was supported by grants to H.O., including a Grant-in-Aid for Scientific Research on Innovative Areas (Foundation of Synapse and Neurocircuit Pathology, 22110001/ 22110002) from the Ministry of Education, Culture, Sports, Science, and Technology of Japan (MEXT), and a Grant-in-Aid for Scientific Research A (16H02655, 19H01042, 22H00464) from the Japanese Society for the Promotion of Science (JSPS).

## Author contributions
J.B.T.: designing genome editing experiments, acquisition and analysis of data, and drafting of the manuscript. Y.Y.: acquisition and analysis of data, and drafting of the manuscript. H.H.: analysis of genome data and drafting of the manuscript. T.T., K.F., M.I., and K.T.: acquisition of data. M.N.: collection of patient history and sample. K.M. and N.M.: iPSC genome data analysis. H.I.: establishment of iPSC. H.T.: conducting iPSC experiments, acquisition and analysis of data, and drafting of the manuscript. H.O.: whole conception and design of the study, analysis of data, obtaining funding, and drafting the manuscript.

## Competing interests
The authors declare no competing interests.
