## [Peer Review File · Communications Medicine]

Reviewers' comments:

Reviewer #1 (Remarks to the Author):

This manuscript describes an AAV2-mediated silencing of PMP22 in CMT1A patient derived iPS cells.

This paper is indeed of scientific interest since this is the first time to show that an AAV serotype is able to transduce a cell line in vitro. AAVs have been shown insufficient to infect monolayer cultures, with only exception being AAV-DRG (which however, has been proven insufficient in vivo). It is very fascinating that co-cultures are transfectable by AAV2. Once this is published, it will be interesting creating the same system for other CMT mutations and AAV serotypes.

Materials and methods and results are well written. However, introduction and discussion need extensive revisit. It gives the impression that authors are new to CMT and gene therapy field. It is mandatory to include recent literature on these fields. Bold scientifically untrue statements were made repeatedly, such as "there is not CMT1A model". There are actually tens of CMT1A mouse models.

For results, it is suggested to show PMP22 mRNA and protein expression in the cell line before and after treatment. Also, to have a solid claim that this is a potential CMT1A therapy, you should show that after AAV2-hSaCas9-gRNAedit infection there is an actual PMP22 silencing by RNA and western blot analysis, not only phenotype improvement.

In the cases when links are used as reference I suggest to change them with peer reviewed papers. In discussion it would be interesting seeing how this method is better than other gene therapy methods already published for CMT1A. Also is there a reason you used AAV2 and not AAV9 or AAV1 that are already in clinical trials?

Genes should be in italics

Specific comments, with recommendations for addressing each comment:

L38: Charcot-Marie-Tooth disease type 1A (CMT1A)

L76: There is extensive literature references for CMT causative genes, I suggest updating the link with peer reviewed papers

L78: and results in a demyelinating peripheral neuropathy

L84: There is extensive literature references for CMT1A frequency, I suggest updating the link with peer reviewed papers

L88: There is extensive literature references for CMT symptoms, I suggest to update this link with peer reviewed papers.

L91: Primary management includes physiotherapy and occupational therapy and in some

L94: There is extensive literature references for CMT treatments/management, I suggest to update this link with peer reviewed papers.

L95: Please clarify why this CMT2B treatment approach is relevant to your work

L99: Replace Sareptas link with NCT number: NCT05011006

L101: There is extensive literature in CMTs gene therapies, from which most of them is for CMT1A, why you don't refer to them? Stavrou et al, 2023, J Peripher Nerv Syst. Might be used as a guide

L103: PMP22 down-regulation is a one of the most promising therapeutic approaches for CMT1A

L104: There are more recent papers showing therapeutic benefit of PMP22 downregulation in CMT1A. Stavrou et al, 2023, J Peripher Nerv Syst. Might be used as a guide

L105-106: This is absolutely not true. There are many mouse models overexpressing human or murine PMP22. All CMT1A mouse models are listed in Stavrou et al Neural Regen Res, 2023

L109: Excessive silencing is NOT causing pathologies similar to PMP22 point mutations (i.e. CMT1E). Excessive silencing is causing pathologies similar to PMP22 deletion i.e HNPP disease

L112: Hence, one of the most promising treatment approaches for CMT1A is genome editing....

L120: leaving the remaining challenge of in vivo delivery in animal models and then humans, which ...

L130-141: I suggest be moved and incorporated in the introduction part. You can refer back to them in results only with a brief sentence.

L147: Is this where you should refer to Fig 1 a and b?

L172: As stated before, this is absolutely not true. There are many CMT1A animal models. Please rephrase as follows: Human iPSC or iPSC-differentiated cells are considered a simple system to test genetic events underlying CMT1A.

L173-175: Moreover.....animal. please remove this sentence.

L195: Need loading control for Fig2 f. All calculations should be done after normalizing with loading control

L238: PMP22 immunostaining (or western blot) is necessary to confirm that co-cultures actually express PMP22 protein (and not only PMP22 mRNA). There are many Schwann cell lines available, most of them express MPZ and S100, but not PMP22. Documenting that (a) what you are trying to silence is actually expressed and (b) you have generated a cell line that expressed PMP22

L302: CMT1A is the most common CMT subtype, accounting for ... (remove PMP22-linked)

L325: This is absolutely untrue. There are many therapeutic approaches directly targeting PMP22. There are many to list them. Stavrou et al, 2023, J Peripher Nerv Syst. Might be used as a guide.

L327: Please add recent literature for CMT1A downregulation

L329: NT-3 and P2X7 are indirect methods that ameliorate CMT1A phenotype while ASO, shRNAs, siRNAs miRs available directly target PMP22 gene and silence it. NT-3 is back on clinical trials (NCT05011006)

L347: Are you sure that multiple AAV injections are feasible to be performed in patients? For a patient to register for a viral gene therapy they need to be first tested negative for neutralizing antibodies against that virus. After a single injection with X virus patients create neutralising antibodies, even if they are immunosuppressed. Multiple injection with the same capsid is not an option for the clinic. Liver toxicity also is analogous with viral concentrations, hence single viral injections are suggested.

L351: There is extensive literature about AAVs injection for peripheral nerve targeting in general and CMTs specifically. Stavrou et al, 2023, J Peripher Nerv Syst. Might be used as a guide.

L359-360: As stated before this is absolutely untrue. There are many CMT1A animal models existing for many decades.

Reviewer #2 (Remarks to the Author):

The manuscript by Taniguchi et al. presents a potential therapeutic approach for Charcot-Marie-Tooth disease type 1A (CMT1A) using genome editing with an AAV2 vector. The researchers developed a method to correct the PMP22 gene duplication based on genome editing with gRNA that targets a unique sequence present at two sites that sandwich only a single copy of duplicated PMP22 genes. They used an AAV2 vector expressing both hSaCas9 and gRNA to infect iPSCs or iPSC-derived Schwann cells, resulting in decreased PMP22 transcript levels by 20 - 40%. The CMT1A-iPSC-derived Schwann cell precursors and Schwann cells exhibited decreased proliferation and increased apoptosis, which extended to impaired myelination in co-culture with normal iPSC-derived neurons. However, the newly-developed AAV2 genome-editing vector normalized the increased apoptosis and impaired myelination ability of human CMT1A-iPSC-derived Schwann cell precursors.

The study provides a promising strategy for treating CMT1A and other genetic disorders caused by gene duplication and highlights the potential of genome editing as a tool for developing effective therapies for inherited diseases.

The gene editing procedure is cleverly designed but needs further validation to be acceptable.

Firstly, more information is needed on the starting iPS line (CiRA00139). Is the duplication in heterozygosity or homozygosity? Has the duplication been finely mapped, and is the karyotype of this iPS line normal before the gene editing?

Secondly, the formula to estimate the efficiency in Figure 2C is an indirect measurement. Direct measurement of editing efficiency can be calculated by screening single clones after the editing using PCR on the genomic DNA followed by Sanger sequencing. This number would give a different level of confidence in the technique. And Sanger sequencing would give a better understanding of the actual genomic changes.

Thirdly, the schematic of the Southern Blot in Figure 2f is not clear. A better schematic, possibly horizontal, should show the position of the enzymes and the probes used. The size and sequence of the probes should be provided.

Fourthly, the ultimate validation of gene editing aimed at correcting a gene duplication is a qPCR showing that the mRNA levels are restored after the correction. I do not find this experiment in the manuscript.

Finally, as for any CRISPR/Cas experiment in iPSC, global and local genomic stability after editing should be tested via karyotype or virtual karyotype. Additionally, off-target editing should be controlled for.

Reviewers' comments:

Reviewer #1 (Remarks to the Author):

This manuscript describes an AAV2-mediated silencing of PMP22 in CMT1A patient derived iPS cells.

This paper is indeed of scientific interest since this is the first time to show that an AAV serotype is able to transduce a cell line in vitro. AAVs have been shown insufficient to infect monolayer cultures, with only exception being AAV-DRG (which however, has been proven insufficient in vivo). It is very fascinating that co-cultures are transfectable by AAV2. Once this is published, it will be interesting creating the same system for other CMT mutations and AAV serotypes.

>>> We appreciate kind efforts of the reviewer for extensive reading of our manuscript and various useful advices.

>>> We appreciate the kind evaluation from the reviewer #1. Of course, we know the classic knowledge that AAVs could not infect a cell line efficiently. However, it is not the case. From around 2004, we expressed genes in neurons by virus vector (Mei-Ling Qi et al, Nature Cell Biol 2007, <https://www.nature.com/articles/ncb1553>), then started the experiments using AAVs (including AAV2 and AAV9) around 2012, and performed a number of experiments infecting AAV to monolayer cultures, in addition to infection of animals including mice and marmosets. Our first paper using AAV was published in EMBO Mol Med (Ito et al, DOI: 10.15252/emmm.201404392) in 2015. The AAV vectors used there was a complex one possessing AAV3 ITRs, AAV2 Rep and AAV1 vp. Since then we have been also using AAV9 and AAV2 vectors. We list examples of our AAV transduction in

the following to show the reviewer that transduction to single layer cells is possible.

1) Mouse primary cortical neuron

Jin et al., Nature Commun 2021 (doi: 10.1038/s41467-021-26851-2.)

2) iPSC-derived neuron

Homma et al., Life Sci Alliance 2021 (doi: 10.26508/lsa.202101022.)

3) iPSC-derived neuron

Tanaka et al., Nature Commun 2020 (doi:

10.1038/s41467-020-14353-6.)

4) Mouse primary cortical neuron

Fujita et al., Sci Rep 2016 (doi: 10.1038/srep31895.)

Materials and methods and results are well written. However, introduction and discussion need extensive revisit. It gives the impression that authors are new to CMT and gene therapy field. It is mandatory to include recent literature on these fields. Bold scientifically untrue statements were made repeatedly, such as “there is not CMT1A model”. There are actually tens of CMT1A mouse models.

>>> We would respond to the comments in details in the following.

>>> First, it is not true that we are new to gene therapy. You can see a number of papers in the following, which bring us a huge amount of experiences. We have also started a project supported by AMED (the biggest governmental funding body for research in Japan) for human clinical trial for SCA1.

- 1) AAV1-pNestin-normal-VCP, AAV1-pNestin-T719A-MCM3 to mutant VCP-KI mice.
Homma et al., Life Sci Alliance 2021 (doi: 10.26508/lsa.202101022.)
- 2) AAV1/2-YAPdeltaC (AAV2 rep and AAV1 rp) to 5xFAD mice.
Tanaka et al., Nature Commun 2020 (doi: 10.1038/s41467-020-14353-6.)
- 3) AAV1-PQBP1 to 5xFAD mice.
Tanaka et al., Mol Psy 2018 (doi: 10.1038/s41380-018-0253-8).
- 4) AAV1/2-RpA1-mCherry (AAV2 rep and AAV1 rp) to mutant Atxn1-KI mice.
Taniguchi et al., Hum Mol Genet 2016 (doi: 10.1093/hmg/ddw272.)
- 5) AAV1-HMGB1, AAV1-HMGB1-EGFP to mutant Atxn1-KI mice.
Ito et al., EMBO Mol Med 2015 (doi: 10.15252/emmm.201404392.)
- 6) AAV1/2-PQBP1 (AAV2 rep and AAV1 rp) to PQBP1 conditional knockout mice.
Ito et al., Mol Psy 2015 (doi: 10.1038/mp.2014.69.)

I am sorry that they are written in Japanese, but I attach the links for our preclinical and clinical projects of gene therapy for SCA1.

<https://research-er.jp/projects/view/965080>

<https://research-er.jp/projects/view/1126587>

>>> Second, it is not true that we are new to CMT though references might have insufficient. One of the authors (Prof. Masanori Nakagawa) is the most experienced and top physician scientist of

CMT in Japan. Please have a look at his publication on PubMed from the link below.

<https://pubmed.ncbi.nlm.nih.gov/?term=Masanori+Nakagawa+Charcot%E2%80%93Marie%E2%80%93Tooth+disease&sort=date>

>>> Third, following the advice, we added a number of recent papers in the field, as you can see in the responses below and in the text.

>>> Fourth, we agree there are tens of CMT1A mouse models. It was very useful for us to read the review paper suggested by the reviewer (Stavrou et al Neural Regen Res, 2023) and to recognize human PMP22 gene-Tg mice (C61 mouse) expressing 2 folds expression of the gene and YAC transgenic mice carrying 560kb genome (8 copies) by the Clare Huxley group. However, they are still transgenic mice of multiple copies randomly integrated in to the genome, and they do not carry duplicated human genome of the PMP22 region sufficiently long to test our method.

The reviewer commented, “Bold scientifically untrue statements were made repeatedly, such as “there is not CMT1A model”.

But we never said anywhere that “there is not CMT1A model” in the previous version.

We said as follows.

L105-106: as no PMP22 gene duplication animal model is available.

L172-173: as no available animal model with a humanized disease genome is currently available.

L359-360: Because no animal model with PMP2 gene duplication is available,

We think these descriptions are true.

However, for avoiding unnecessary discussion, we added or changed description about mouse models in multiple places of the text, to clarify how we think regarding CMT1A mouse models.

Meanwhile since our knowledge is limited, and we have not used ChatGPT or IBM-Watson for searching all the publications in all languages for the search of CMT1A mouse models, so if the reviewer knows the knock-in mouse model exactly possessing human CMT gene duplication with sufficient genome regions, we are extremely glad to include that paper and correct our description.

For results, it is suggested to show PMP22 mRNA and protein expression in the cell line before and after treatment.

Also, to have a solid claim that this is a potential CMT1A therapy, you should show that after AAV2-hSaCas9-gRNAedit infection there is an actual PMP22 silencing by RNA and western blot analysis, not only phenotype improvement.

>>> We appreciate the critical comments. We performed the requested experiments and showed the results in new Figure 7.

In the cases when links are used as reference I suggest to change them with peer reviewed papers.

>>> We followed the advice and added recent review papers.

In discussion it would be interesting seeing how this method is better than other gene therapy methods already published for CMT1A. Also is there a reason you used AAV2 and not AAV9 or AAV1 that are already in clinical trials?

>>> As the reviewer pointed out AAV9 and AAV1 are becoming popular. However, AAV2 and the derivatives have a long history including human clinical trials (in this sense, the comment is wrong).

AAV2 has been used in human patients. The following are a few examples.

- Rocco MT, Akhter AS, Ehrlich DJ, Scott GC, Lungu C, Munjal V, Aquino A, Lonser RR, Fiandaca MS, Hallett M, Heiss JD, Bankiewicz KS. Long-term safety of MRI-guided administration of AAV2-GDNF and gadoteridol in the putamen of individuals with Parkinson's disease. *Mol Ther*. 2022 Dec 7;30(12):3632-3638. doi: 10.1016/j.ymthe.2022.08.003. Epub 2022
- Aleman TS, Huckfeldt RM, Serrano LW, Pearson DJ, Vergilio GK, McCague S, Marshall KA, Ashtari M, Doan TM, Weigel-DiFranco CA, Biron BS, Wen XH, Chung DC, Liu E, Ferenchak K, Morgan JIW, Pierce EA, Elliott D, Bennett J, Comander J, Maguire AM. Adeno-Associated Virus Serotype 2-hCHM Subretinal Delivery to the Macula in Choroideremia: Two-Year Interim Results of an Ongoing Phase I/II Gene Therapy Trial. *Ophthalmology*. 2022 Oct;129(10):1177-1191.
- George LA, Ragni MV, Rasko JEJ, Raffini LJ, Samelson-Jones BJ, Ozelo M, Hazbon M, Runowski AR, Wellman JA, Wachtel K, Chen Y, Anguela XM, Kuranda K, Mingozi F, High KA. Long-Term Follow-Up of the First in Human Intravascular Delivery of AAV for Gene Transfer: AAV2-hFIX16 for Severe Hemophilia B. *Mol Ther*. 2020 Sep 2;28(9):2073-2082.

Also in the reviews you recommended (Stavrou et al, 2021; 2023), there are cases using AAV2 for CMT1A.

- Serfecz J, Bazick H, Al Salihi MO, Turner P, Fields C, Cruz P, Renne R, Notterpek L. Downregulation of the human peripheral myelin protein 22 gene by miR-29a in cellular models of Charcot-Marie-Tooth disease. *Gene Ther.* 2019 Dec;26(12):455-464. doi: 10.1038/s41434-019-0098-z.

Genes should be in italics

>>> Thank you for the advice. We corrected them.

Specific comments, with recommendations for addressing each comment:

L38: Charcot-Marie-Tooth disease type 1A (CMT1A)

>>> We corrected it.

L76: There is extensive literature references for CMT causative genes, I suggest updating the link with peer reviewed papers

>>> We added new peer reviewed papers including the critical and most updated ones like GeneReviews® [Internet] with web links.

L78: and results in a demyelinating peripheral neuropathy

>>> New references abovementioned include demyelinating peripheral neuropathies.

L84: There is extensive literature references for CMT1A frequency, I suggest updating the link with peer reviewed papers

>>> We added new peer reviewed papers including the critical and most updated ones.

L88: There is extensive literature references for CMT symptoms, I suggest to update this link with peer reviewed papers.

>>> We added new peer reviewed papers including the critical and most updated ones.

L91: Primary management includes physiotherapy and occupational therapy and in some

>>> We thank the advice, and followed it.

L94: There is extensive literature references for CMT treatments/management, I suggest to update this link with peer reviewed papers.

>>> We added new peer reviewed papers.

L95: Please clarify why this CMT2B treatment approach is relevant to your work

>>> We delete this part.

L99: Replace Sareptas link with NCT number: NCT05011006

>>> It is possible to use <https://clinicaltrials.gov/study/NCT05011006> for NCT05011006. However, the clinical trial record does not include the plan of the company, and here we want to show that the company has a plan of clinical trials. So the meaning is changes if we refer NCT05011006.

>>> But, as we delete the description about CMT2B treatment approach, the link became unnecessary and also disappeared.

L101: There is extensive literature in CMTs gene therapies, from which most of them is for CMT1A, why you don't refer to them? Stavrou et al, 2023, J Peripher Nerv Syst. Might be used as a guide

>>> We added recent papers relevant to gene therapies and ASO. Thank you for the advice about the review paper, it was very useful.

L103: PMP22 down-regulation is a one of the most promising therapeutic approaches for CMT1A

>>> We corrected the sentence following the advice.

L104: There are more recent papers showing therapeutic benefit of PMP22 downregulation in CMT1A. Stavrou et al, 2023, J Peripher Nerv Syst. Might be used as a guide

>>> We thank the reviewer #1 for suggesting a nice review paper.

L105-106: This is absolutely not true. There are many mouse models overexpressing human or murine PMP22. All CMT1A mouse models are listed in Stavrou et al Neural Regen Res, 2023

>>> We agree there are many animal models. But no model mice possess human genome more than 1.5Mb in which PMP22 genes are aligned in tandem. To avoid confusion, we changed expression and explained our idea more in details.

L109: Excessive silencing is NOT causing pathologies similar to PMP22 point mutations (i.e. CMT1E). Excessive silencing is causing pathologies similar to PMP22 deletion i.e HNPP disease

>>> We thank the advice and deleted it.

L112: Hence, one of the most promising treatment approaches for CMT1A is genome editing....

>>> We corrected the sentence following the advice.

L120: leaving the remaining challenge of in vivo delivery in animal models and then humans, which ...

>>> We corrected the sentence following the advice.

L130-141: I suggest be moved and incorporated in the introduction part. You can refer back to them in results only with a brief sentence.

>>> We followed the advice and moved this section to Introduction.

L147: Is this where you should refer to Fig 1 a and b?

>>> L147 is the position to refer Supplementary Figure 1a, so we referred them. We referred Figure 1 and Supplementary Figure 1 at appropriate positions.

L172: As stated before, this is absolutely not true. There are many CMT1A animal models. Please rephrase as follows: Human iPSC or iPSC-differentiated cells are considered a simple system to test genetic events underlying CMT1A.

>>> There are many animal models (we agree), but no humanized genome model with duplicated human PMP22 genome in the original alignment and with sufficiently long flanking regions of human genome around PMP22 gene to examine our method.

>>> We also used the phrase suggested by the reviewer #1.

L173-175: Moreover.....animal. please remove this sentence.

>>> We think this is one of the important reasons for why we used human iPSC (but not 560kb YAC Tg mice, for instance).

L195: Need loading control for Fig2 f. All calculations should be done after normalizing with loading control

>>> We added the loading control in Fig 2f.

L238: PMP22 immunostaining (or western blot) is necessary to confirm that co-cultures actually express PMP22 protein (and not only PMP22 mRNA). There are many Schwann cell lines available, most of them express MPZ and S100, but not PMP22. Documenting that (a) what you are trying to silence is actually expressed and (b) you have generated a cell line that expressed PMP22

>>> We performed the requested experiment and showed the result in Figure 7.

L302: CMT1A is the most common CMT subtype, accounting for ...
(remove PMP22-linked)

>>> We removed "PMP22-linked".

L325: This is absolutely untrue. There are many therapeutic approaches directly targeting PMP22. There are many to list them. Stavrou et al, 2023, J Peripher Nerv Syst. Might be used as a guide.

>>> Stavrou et al, 2023, J Peripher Nerv Syst had been already referred as ref 35 in previous version (now the ref number is changes by the increase of references). We changed the expression in L325 to prevent misunderstanding of readers, and added important papers in regards of therapy. Additionally we referred this paper by describing that Stavrou et al summarizes current therapeutics including gene therapies.

L327: Please add recent literature for CMT1A downregulation

>>> We added Stavrou et al, JCI 2022.

L329: NT-3 and P2X7 are indirect methods that ameliorate CMT1A phenotype while ASO, shRNAs, siRNAs miRs available directly target PMP22 gene and silence it. NT-3 is back on clinical trials (NCT05011006)

>>> We added papers regarding direct approaches to target PMP22.

L347: Are you sure that multiple AAV injections are feasible to be performed in patients? For a patient to register for a viral gene

therapy they need to be first tested negative for neutralizing antibodies against that virus. After a single injection with X virus patients create neutralising antibodies, even if they are immunosuppressed. Multiple injection with the same capsid is not an option for the clinic. Liver toxicity also is analogous with viral concentrations, hence single viral injections are suggested.

>>> We agree with the comment. That is why we wrote “However, an overly effective vector would not be appropriate for early clinical trials, as this could potentially cause unexpected side effects.” in L347-349 of previous version.

We changed the expression in sentences of this section to prevent misunderstanding of readers.

L351: There is extensive literature about AAVs injection for peripheral nerve targeting in general and CMTs specifically. Stavrou et al, 2023, J Peripher Nerv Syst. Might be used as a guide.

>>> Stavrou et al, 2023, J Peripher Nerv Syst is a good review, we agree. We referred this paper by describing that Stavrou et al summarizes current therapeutics including gene therapies in other place of the paper, and also added many important papers in regards of gene therapy in multiple places of the new version.

L359-360: As stated before this is absolutely untrue. There are many CMT1A animal models existing for many decades.

>>> In L359-360 of previous version, we described that “Because no animal model with PMP22 gene duplication is available,”. We think it is true. If the reviewer knows mouse models carrying human

duplicated PMP22 genes located in tandem in 4 Mb genome region,
please tell us.

Reviewer #2 (Remarks to the Author):

The manuscript by Taniguchi et al. presents a potential therapeutic approach for Charcot-Marie-Tooth disease type 1A (CMT1A) using genome editing with an AAV2 vector. The researchers developed a method to correct the PMP22 gene duplication based on genome editing with gRNA that targets a unique sequence present at two sites that sandwich only a single copy of duplicated PMP22 genes. They used an AAV2 vector expressing both hSaCas9 and gRNA to infect iPSCs or iPSC-derived Schwann cells, resulting in decreased PMP22 transcript levels by 20 - 40%. The CMT1A-iPSC-derived Schwann cell precursors and Schwann cells exhibited decreased proliferation and increased apoptosis, which extended to impaired myelination in co-culture with normal iPSC-derived neurons. However, the newly-developed AAV2 genome-editing vector normalized the increased apoptosis and impaired myelination ability of human CMT1A-iPSC-derived Schwann cell precursors.

The study provides a promising strategy for treating CMT1A and other genetic disorders caused by gene duplication and highlights the potential of genome editing as a tool for developing effective therapies for inherited diseases.

>>> We appreciate very much kind evaluation of the reviewer #2.

The gene editing procedure is cleverly designed but needs further validation to be acceptable.

Firstly, more information is needed on the starting iPS line (CiRA00139). Is the duplication in heterozygosity or homozygosity?

>> Duplication is heterozygous judging from the family history and also from 1.5 folds increase of reads in whole genome sequencing data (Sup Fig 2).

Has the duplication been finely mapped,

>> We performed whole genome sequencing and mapped duplicated genome region by the change of reads.

and is the karyotype of this iPS line normal before the gene editing?

>> Karyotype of this iPS line was examined and confirmed normal. We showed the result in Sup Fig 2.

Secondly, the formula to estimate the efficiency in Figure 2C is an indirect measurement. Direct measurement of editing efficiency can be calculated by screening single clones after the editing using PCR on the genomic DNA followed by Sanger sequencing. This number

would give a different level of confidence in the technique. And Sanger sequencing would give a better understanding of the actual genomic changes.

>>> Regarding the method and the size of analysis, we consulted with the editor. The editor agreed our proposal, so we performed the “TOPO subcloning and Sanger sequencing of plasmids”. This is probably what the reviewer #2 might have suggested by “PCR on the genomic DNA followed by Sanger sequencing”.

We need to say, however, efficiency of editing was estimated rather higher in this method. It is because our qPCR and Southern blot methods estimated the efficiency of “genome editing + cleave out of deletion of 1.5Mb genome” (only successful cases), while plasmid sequencing estimated the efficiency of genome editing (including unsuccessful cases like re-ligation without deletion of 1.5Mb genome).

Thirdly, the schematic of the Southern Blot in Figure 2f is not clear. A better schematic, possibly horizontal, should show the position of the enzymes and the probes used. The size and sequence of the probes should be provided.

>>> We made the scheme of Figure 2f horizontal. We also described position, size and sequence of the probes in Figure 2.

Fourthly, the ultimate validation of gene editing aimed at correcting a gene duplication is a qPCR showing that the mRNA levels are restored after the correction. I do not find this experiment in the manuscript.

>>> We confirmed the reduction of the mRNA to normal level by genome editing in new Figure 7.

Finally, as for any CRISPR/Cas experiment in iPSC, global and local genomic stability after editing should be tested via karyotype or virtual karyotype. Additionally, off-target editing should be controlled for.

>>> We examined karyotype of iPSC before and after genome editing.

>>> We performed WGS of iPSC before and after editing to evaluate off-target effects. We also performed intraneural injection to sural nerve and examined whole genome sequencing of the tissues including nerve and surrounding muscles.

In both cases, off-target mutations were quite few in comparison to the whole genome, while further improvement should be achieved by technical advances in genome editing.

Reviewers' comments:

Reviewer #1 (Remarks to the Author):

The authors have addressed all the comments and had performed extra experiments to strengthen their paper.

To me Fig.7 western blot image is not reflecting silencing quantification shown in the graph. However, I think that the paper should proceed with publication even without improving this figure.

Reviewer #2 (Remarks to the Author):

The revised manuscript presents the main findings in an improved manner, and it properly addresses my concerns except for my point on the quantification of editing efficiency.

Once again, I'd like to emphasize that this is not a critical issue. The story remains valuable for publication even without the precise quantification of cells that have undergone editing. Nevertheless, if the authors intend to provide a quantification, it should be executed meticulously.

I maintain the perspective that asserting a precise quantification of editing efficiency based on bulk data, be it through qPCR (Fig. 2C) or Southern blotting (Fig. 2G), is not possible. The detection of a one-fold increase is exceedingly challenging in qPCR, even within a single clone, let alone within a larger population. Furthermore, the intensity of bands in Southern blotting does not accurately correlate with the count of edited genomes. The efforts shown in Figure S4 are similarly non-quantifiable, as the authors acknowledge that the assay only identifies successful instances.

The authors posit that the editing efficiency is estimated to be "rather high." As a result, I propose the execution of a Southern blot analysis on a minimum of 20 individual iPSC clones, followed by a tally of clones that have been successfully edited (i.e., those lacking the hybrid 3.3kb band).

I commend the inclusion of WGS analysis, and the observation that only a few changes were indeed detected is reassuring. For the sake of comprehensive reporting, it would be valuable to determine the percentage of detected changes situated within coding regions or regulatory domains.

Reviewers' comments:

Reviewer #1 (Remarks to the Author):

The authors have addressed all the comments and had performed extra experiments to strengthen their paper.

>>> Thank you very much for kind evaluation.

To me Fig.7 western blot image is not reflecting silencing quantification shown in the graph. However, I think that the paper should proceed with publication even without improving this figure.

>>> Following the advice of reviewer #1, we replaced the representative images.

Reviewer #2 (Remarks to the Author):

The revised manuscript presents the main findings in an improved manner, and it properly addresses my concerns except for my point on the quantification of editing efficiency.

>>> Thank you very much for kind evaluation.

Once again, I'd like to emphasize that this is not a critical issue. The story remains valuable for publication even without the precise quantification of cells that have undergone editing. Nevertheless, if the authors intend to provide a quantification, it should be executed meticulously.

I maintain the perspective that asserting a precise quantification of editing

efficiency based on bulk data, be it through qPCR (Fig. 2C) or Southern blotting (Fig. 2G), is not possible. The detection of a one-fold increase is exceedingly challenging in qPCR, even within a single clone, let alone within a larger population. Furthermore, the intensity of bands in Southern blotting does not accurately correlate with the count of edited genomes. ===The efforts shown in Figure S4 are similarly non-quantifiable, as the authors acknowledge that the assay only identifies successful instances.===

The authors posit that the editing efficiency is estimated to be "rather high." As a result, I propose the execution of a Southern blot analysis on a minimum of 20 individual iPSC clones, followed by a tally of clones that have been successfully edited (i.e., those lacking the hybrid 3.3kb band).

>>> We are sorry, but we are afraid that this inquiry for a new experiment is not rational scientifically. The reviewer #2 previously asked us to perform Sanger sequencing of PCR products from a large number of cell clones. We actually performed Sanger sequencing of PCR products from 1,000 clones. Moreover, we analyzed the editing efficiency by Southern blot with DNA prepared from 2×10^6 cells. This time, the reviewer #2 demands us to perform Southern blot with DNA prepared from 20 cell clones of AAV-edited CMT-iPSC. Though it might erase 3.3kb band in a part of cells, the estimated editing efficiency with 20 cells is unreliable (even if it were 100 cells) due to the stochastic variance with a very small number (20 cells \lll 2×10^6 cells). Moreover, the experiments with 20 CMT-iPSC clonal lines takes more than 2 months, just for such an unreliable data. If with 100 clones, it becomes impossible.

>>> Regarding the claim

===The efforts shown in Figure S4 are similarly non-quantifiable, as the authors acknowledge that the assay only identifies successful instances.===

>>> This is misunderstanding. We excluded the cases amplified from unrelated genome regions. But we did not artificially select successful cases.

>>> We therefore consulted with the editor and received the answer from the editor that we need not do this new experiment but we should describe concerns for incompleteness of qPCR and Southern blot in the text.

I commend the inclusion of WGS analysis, and the observation that only a few changes were indeed detected is reassuring. For the sake of comprehensive reporting, it would be valuable to determine the percentage of detected changes situated within coding regions or regulatory domains.

>>> Thank you for kind evaluation of our WGS analysis. Following the advice we added the number (percentage) of detected changes within coding/regulatory regions in Supplementary Figure 8 and in text. For instance, in human iPSC after genome editing, one SNP was located in coding region, but deduced amino acid sequence was not changed from original one in FCGBP gene as shown below. CGC(Arg)→CGA(Arg).

We calculated the percentage (or frequency) of SNP that might be induced by genome editing in the whole genome.